# 🐯TIGeR: Unifying Text-to-Image Generation and Retrieval with Large Multimodal Models

**Leigang Qu**[1]  **Haochuan Li**[1]  **Tan Wang**[2*]  **Wenjie Wang**[3*]  **Yongqi Li**[4]
**Liqiang Nie**[5]  **Tat-Seng Chua**[1]
[1]National University of Singapore  [2]Nanyang Technological University
[3]University of Science and Technology of China  [4]Hong Kong Polytechnic University
[5]Harbin Institute of Technology (Shenzhen)
{leigangqu, wenjiewang96, nieliqiang, liyongqi0}@gmail.com
haochuan@u.nus.edu   tan317@ntu.edu.sg   dcscts@nus.edu.sg

## Abstract

How humans can effectively and efficiently acquire images has always been a perennial question. A classic solution is *text-to-image retrieval* from an existing database; however, the limited database typically lacks creativity. By contrast, recent breakthroughs in *text-to-image generation* have made it possible to produce attractive and counterfactual visual content, but it faces challenges in synthesizing knowledge-intensive images. In this work, we rethink the relationship between text-to-image generation and retrieval, proposing a *unified* framework for both tasks with one single Large Multimodal Model (LMM). Specifically, we first explore the intrinsic discriminative abilities of LMMs and introduce an efficient generative retrieval method for text-to-image retrieval in a training-free manner. Subsequently, we unify generation and retrieval autoregressively and propose an autonomous decision mechanism to choose the best-matched one between generated and retrieved images as the response to the text prompt. To standardize the evaluation of unified text-to-image generation and retrieval, we construct TIGeR-Bench, a benchmark spanning both creative and knowledge-intensive domains. Extensive experiments on TIGeR-Bench and two retrieval benchmarks, *i.e.*, Flickr30K and MS-COCO, demonstrate the superiority of our proposed framework. The code, models, and benchmark are available at `https://tiger-t2i.github.io`.

## 1 Introduction

The explosion of visual information on the Web significantly challenges human information access. Text-to-Image Retrieval (T2I-R) (Radford et al., 2021; Yu et al., 2022; Li et al., 2023b) is one of the main channels to obtain visual information given a text prompt. However, T2I-R is limited to retrieving existing images in the database, lacking flexibility and creativity. Furthermore, as the database expands, retrieval costs increase significantly. Recent years have witnessed thrilling progress in Text-to-Image Generation (T2I-G) (Ramesh et al., 2022; Rombach et al., 2022; Podell et al., 2023), which directly generates new images to meet human visual information needs. However, T2I-G struggles with knowledge-intensive concepts such as landmarks and natural species (see the right part of Fig. 1), often resulting in hallucination issues (Kim et al., 2024; Huang et al., 2024b). In this light, a single T2I-R or T2I-G model may not satisfy the diverse and evolving human information needs. It is pivotal to unify both T2I-R and T2I-G within a framework for visual information delivery.

To this end, a straightforward solution is to empower discriminative models with the generation ability. However, the early-stage trial (*e.g.*, JEM (Grathwohl et al., 2019)) requires extra generative training and may compromise the original discriminative power. Another solution adapts generative models such as diffusion models (Podell et al., 2023) to achieve the discriminative tasks (Li et al., 2023a; Clark & Jaini, 2023). Despite the significance, these methods are limited to diffusion models and inevitably suffer from the notorious inefficiency problem caused by iterative denoising (Ho et al.,

---

*Corresponding Authors: Tan Wang and Wenjie Wang

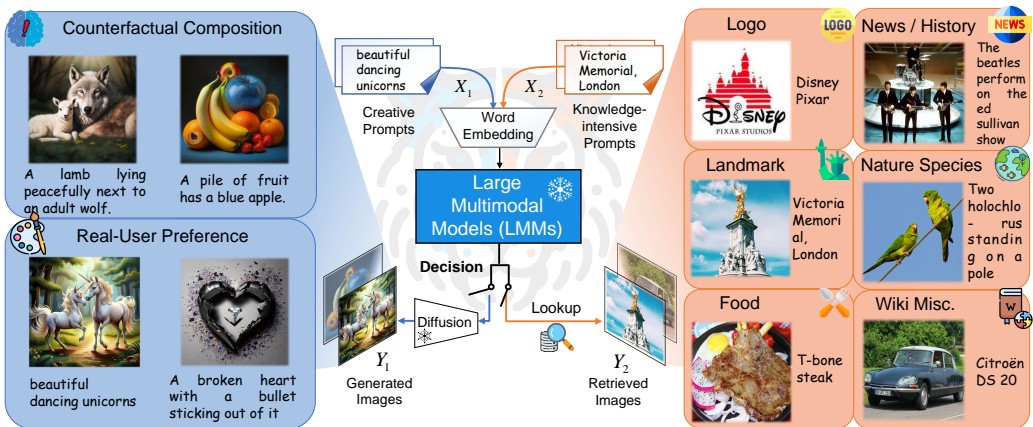

Figure 1: **TIGeR-ONE** unifies T2I-G and T2I-R through one single LMM in a training-free autoregressive way, with a decision mechanism to adaptively select between generated and retrieved images based on user prompts. Besides, we construct TIGeR-Bench, encompassing eight creative and knowledge-intensive domains in total to facilitate a comprehensive evaluation of TIGeR.

2020). Moreover, diffusion models are usually tailored for simple discriminative tasks such as image classification (Li et al., 2023a; He et al., 2023) and making them less suitable for processing complex human prompts for large-scale retrieval in practical scenarios (Schuhmann et al., 2022).

Unlike diffusion models, Large Multimodal Models (LMMs) offer another form of generative paradigm to solve broad vision-language problems, garnering significant attention for their powerful language understanding and instruction following abilities (Ouyang et al., 2022; Touvron et al., 2023). Recently, notable efforts in LMMs (Sun et al., 2023a; Ge et al., 2023; Dong et al., 2023) integrate Large Language Models (LLMs) with external T2I-G models (Rombach et al., 2022) for image synthesis. However, most studies focus solely on T2I-G, neglecting T2I-R. Even though GILL (Koh et al., 2023) fuses an LLM and the image encoder & decoder to enable both generation and retrieval in an ensemble strategy, it essentially incorporates an external retriever (*i.e.*, CLIP (Radford et al., 2021)) for dense retrieval in a dedicated embedding space. As such, GILL brings extra alignment costs and still suffers from the inefficiency problem (discussed in Sec. 2 and Sec. 5.3) of the dense retrieval paradigm, particularly in large-scale image retrieval scenarios (Tay et al., 2022).

To this end, we propose to unify Text-to-Image Generation and Retrieval (**TIGeR**) in this work, and present a model-agnostic framework named **TIGeR-ONE** that achieves this unification within one single LMM, enabling flexible and efficient image acquisition as shown in Fig. 1. We first delve into the intrinsic bidirectional (*i.e.*, text-to-image and image-to-text) discriminative abilities of LMMs in Sec. 3.2. Specifically, we investigate three likelihood-based proxies to estimate cross-modal semantic similarities. Based on these proxies, TIGeR-ONE adopts an efficient generative retrieval method with forward beam search and reverse re-ranking as in Sec. 3.3, unifying both T2I-R and T2I-G in an autoregressive generation manner. Moreover, TIGeR-ONE presents an autonomous decision mechanism to adaptively select between retrieved and generated images based on user prompts.

Existing benchmarks (Saharia et al., 2022; Huang et al., 2023; Weyand et al., 2020) assess generation and retrieval separately, with limited domain coverage. To comprehensively evaluate the performance of LMMs on TIGeR, we build a benchmark called **TIGeR-Bench** (Sec. 4). It encompasses creative images from the counterfactual world and imaginative scenarios (Kirstain et al., 2024), and knowledge-intensive images from six diverse domains (*e.g.*, logo, landmark, and natural species). We carry out extensive experiments to assess the TIGeR performance of representative LMMs on TIGeR-Bench and two T2I-R benchmarks, validating the effectiveness and efficiency of TIGeR-ONE. Overall, we summarize the contributions into three points.

- Driven by the complementary roles of text-to-image generation and retrieval in visual information access, we propose unifying both tasks to meet complex human information needs.

- We comprehensively inspect the intrinsic cross-modal discriminative abilities of LMMs and propose TIGeR-ONE, a model-agnostic framework for the TIGeR task. TIGeR-ONE performs text-to-image generation and retrieval in a training-free autoregressive manner, selecting the best-matched result autonomously and efficiently.

- We construct a comprehensive image acquisition benchmark, TIGeR-Bench, to evaluate the performance of TIGeR on LMMs in creative and knowledge-intensive domains. Extensive experiments on TIGeR-Bench and two T2I-G benchmarks including Flickr30K (Young et al., 2014) and MS-COCO (Lin et al., 2014) verify the effectiveness of TIGeR-ONE.

## 2 RELATED WORK

**Text-to-Image Generation.** T2I-G has aroused wide attention in both academia and industry over the past decade, with advancements ranging from Generative Adversarial Networks (Reed et al., 2016) to Auto-regression Models (Ding et al., 2021) and Diffusion Probabilistic Models (DPMs) (Ho et al., 2020). Recent breakthroughs in DPMs, guided by the scaling law (Kaplan et al., 2020; Li et al., 2024a), have propelled T2I-G to new heights, *e.g.*, models like DALL-E 2 (Ramesh et al., 2022) and DALL-E 3 (Betker et al., 2023), the Imagen series (Saharia et al., 2022), and the Stable Diffusion (SD) series (Rombach et al., 2022; Podell et al., 2023; Esser et al., 2024). Concurrently, efforts have been made to enhance the composed text-image alignment (Feng et al., 2022; Chefer et al., 2023; Qu et al., 2024; Yang et al., 2024) and cater to human preference (Lee et al., 2023; Xu et al., 2024). Some studies recognize the importance of knowledge-intensive image acquisition, employing RAG (Chen et al., 2022) or constructing benchmark (Huang et al., 2024a). However, they focus solely on generation and overlook the potential for unifying generation and retrieval.

**Text-to-Image Retrieval.** Early studies on multimodal information retrieval focused on feature representation (Faghri et al., 2017; Li et al., 2019) and modality interaction (Lee et al., 2018; Qu et al., 2021) for precise cross-modal similarity estimation. Recent advancements, propelled by large-scale pre-training, have led to improved retrieval performance and generalization in vision-language models (Radford et al., 2021; Li et al., 2021; Dou et al., 2022). More recently, researchers have explored more challenging scenarios, such as fine-grained interaction (Lin et al., 2024a), equivariant similarity (Wang et al., 2023), multimodal instruction following (Wei et al., 2023), and chat-based IR (Levy et al., 2024). Despite thrilling progress, retrieval systems are inherently limited by database size, and incapable of creating new visual content.

**Large Multimodal Models.** Empowered by the versatility of LLMs, pioneering works on LMMs have shown impressive understanding capabilities (Liu et al., 2023; Zhu et al., 2023a). Recent research explores image generation through two categories: 1) *Continuous Visual Representation* methods aim to align visual representations from LLMs with condition embeddings of SD through regression (Koh et al., 2023; Sun et al., 2023a; Wu et al., 2023; Dong et al., 2023; Zhu et al., 2023b) or score distillation (Dong et al., 2023) objectives. GILL (Koh et al., 2023) combines an external dense retriever, *i.e.*, CLIP, and a diffusion decoder with an LLM to achieve retrieval and generation, respectively. However, such an ensemble approach brings extra alignment costs and may encounter an alignment gap (Zhao et al., 2024). Moreover, the dense retriever suffers from inefficiency (Li et al., 2024b), as it requires extensive similarity comparisons between the query and all items in the database. 2) *Discrete Visual Tokenization* methods (Yu et al., 2023; Lu et al., 2023; Ge et al., 2023) first encode an image into a sequence of discrete codes (Esser et al., 2021; Van Den Oord et al., 2017; Ge et al., 2023; Jin et al., 2023), and then employ next-token prediction to train LMMs. To synthesize images, the discrete codes are decoded into the pixel space via VQ-GAN or SD. In this work, we resort to the discrete paradigm to be consistent with the inherent discreteness of language. Compared with existing work, TIGeR-ONE achieves comprehensive image acquisition, encompassing both content creation and knowledge retrieval within a single framework.

## 3 METHODOLOGY

We first formulate the task of unifying T2I-G and T2I-R through LMMs in Sec. 3.1. We then explore the intrinsic cross-model discriminative ability of LMMs in Sec. 3.2. Based on the discriminative ability, we propose the TIGeR-ONE framework, as shown in Fig. 2, including generative retrieval in Sec. 3.3, synchronous generation and retrieval, and decision-making in Sec. 3.4.

## 3.1 TASK FORMULATION

TIGeR aims to satisfy complex human visual information needs by unifying T2I-G and T2I-R in a unified LMM framework. We formulate this problem in an autoregressive generation manner:

$$p(Y|X) = \prod_{i=1}^{N} p(y_i|Y_{<i}, X), \tag{1}$$

where $X$ denotes a textual prompt provided by humans, tokenized into a sequence $X = [x_1, ..., x_M]$; and $Y = [y_1, ..., y_N]$ denotes the sequence of visual tokens (Ge et al., 2023) that can be decoded into an image, with $Y_{<i}$ referring to the tokens before step $i$. By sampling from the conditional distribution, we obtain a sequence instance, *i.e.*, $Y^* \sim p(Y|X)$, where $Y^* \in \mathbb{V}^N$ and $\mathbb{V}$ denotes the visual token space defined by a visual vocabulary, and $|\mathbb{V}| = V$ is the vocabulary size. $\mathbb{V}^N$ denotes the Cartesian product of $N$ token spaces, *i.e.*, the whole discrete visual space.

To achieve unified T2I-G and T2I-R, an LMM is required to possess the following three capabilities: 1) **creativity** to generate a novel photorealistic image $\hat{Y}$ based on the visual tokens sampled from $p(Y|X)$; 2) **discrimination** to measure semantic similarity between a prompt $X$ and each image $Y$, and then retrieve the relevant image $\tilde{Y}$ from a database $\mathcal{G}$, formally, $\tilde{Y} = \arg\max_{Y \in \mathcal{G}} p(Y|X)$; and 3) **decision** to automatically determine the superior option by comparing $p(\hat{Y}|X)$ and $p(\tilde{Y}|X)$ for generation and retrieval, respectively, ultimately yielding the optimal result $Y^*$.

Considering the proficiency of recent LMMs (Dong et al., 2023; Zheng et al., 2023; Ge et al., 2023) in creative T2I-G, we shed more light on exploring the discriminative ability for T2I-R and the potential of unifying generation and retrieval with decision-making in the remaining of this section.

## 3.2 INTRINSIC DISCRIMINATIVE ABILITY OF LMMs

For effective T2I-R, we first probe the discriminative capability of LLMs and present three training-free proxies to estimate the semantic similarity between the prompt and images in the database.

**Proxy 1: Conditional Likelihood**. To estimate the semantic similarity $s(X, Y)$ between a given text prompt $X$ and an image $Y \in \mathcal{G}$, a straightforward approach is to employ the conditional likelihood based on autoregressive factorization as the proxy:

Table 1: Text-to-image ranking performance of three similarity (Sim.) proxies for SEED-LLaMA (Ge et al., 2023) and LaVIT (Jin et al., 2023) on MS-COCO (Lin et al., 2014).

$$s(X, Y) = \log p(Y|X) = \sum_{i=1}^{N} \log p(y_i|Y_{<i}, X), \tag{2}$$

| Sim. Proxy | SEED-LLaMA | | LaVIT | |
|---|---|---|---|---|
| | R@1 | R@5 | R@1 | R@5 |
| Random | 0.02 | 0.10 | 0.02 | 0.10 |
| $\log p(Y\|X)$ | 3.50 | 8.79 | 0.02 | 0.16 |
| $\log \frac{p(Y\|X)}{p(Y)}$ | 26.25 | 54.57 | 23.43 | 48.52 |
| $\log p(X\|Y)$ | 49.34 | 75.45 | 50.35 | 70.87 |

where $p(Y|X)$ denotes the likelihood of autoregressively generating $Y$ conditioned on the given $X$. In practice, we can attain it by computing the cross entropy between the predicted logits and the image tokens. However, as shown in Tab. 1, we observed that this proxy performs poorly. We attribute this issue to *visual bias*, caused by the interference of the visual prior $p(Y)$. Although similar phenomena have been noted in recent studies (Krojer et al., 2023; Lin et al., 2024b) on diffusion and captioning models, the visual bias problem in LMMs has yet to receive adequate research attention.

**Proxy 2: Debiasing Pointwise Mutual Information**. In this study, the visual bias largely stems from the unbalanced distribution $p(Y)$, and thus we introduce an alternative proxy based on Pointwise Mutual Information (PMI) (Role & Nadif, 2011; Li et al., 2015; Lin et al., 2024b) as,

$$s(X, Y) = \log \frac{p(Y|X)}{p(Y)^\eta} = \sum_{i=1}^{N} \log p(y_i|Y_{<i}, X) - \eta \sum_{i=1}^{N} \log p(y_i|Y_{<i}), \tag{3}$$

where we use the visual prior $p(Y)$ to help debiasing with a strength factor $\eta$. $p(Y)$ can be approximately estimated by $p(Y|\bar{X})$, where $\bar{X}$ refers to a special prompt without any descriptive content, *e.g.*, a null character or "Can you give me an image?". The results in Tab. 1 demonstrate that this proxy could significantly alleviate the visual bias issue.

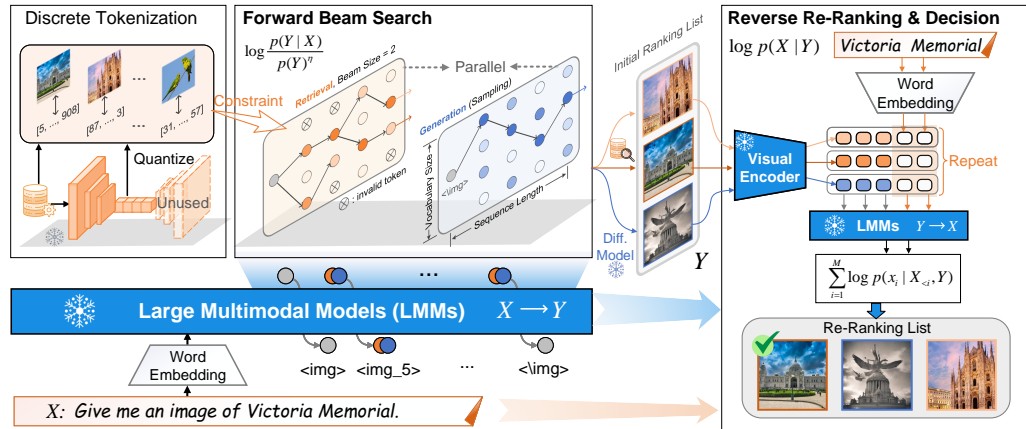

Figure 2: Overview of the TIGeR-ONE framework to unify text-to-image generation and retrieval. Images from the database are first tokenized into discrete codes and a lookup table is maintained for the correspondence between discrete codes and images. The given prompt $X$ is first fed into an LMM and Forward Beam Search is performed to retrieve and generate images in parallel. The prompt and obtained images are then fed into the same LLM for Reverse Re-Ranking and Decision-making.

**Proxy 3: Reverse Conditional Likelihood**. In addition to the debiasing strategy, we propose another option to circumvent the unbalanced prior distribution. Different from the generation process without the image $Y$, the retrieval task allows the model to access all images in the database. This means we can estimate the semantic similarity in a reverse way by predicting the conditional likelihood of $X$ given $Y$ to alleviate the visual bias of T2I-G models. Formally,

$$s(X, Y) = \log p(X|Y) = \sum_{i=1}^{M} \log p(x_i|X_{<i}, Y), \tag{4}$$

where the LMMs work as image captioners to estimate the semantic similarity. As shown in Tab. 1, this reverse proxy outperforms others, effectively revealing the discriminative abilities of LMMs.

Now we could estimate similarities between a given prompt and all $|\mathcal{G}|$ images in the database by traversing each image and calculating a proxy. Afterward, we could sort them to attain a ranking list, enabling T2I-R. Compared with the prior work GILL (Koh et al., 2023), any of the three proxies can be seamlessly integrated with next-token prediction, the mainstream paradigm of generative pre-training. This proxy-based approach eliminates the need for additional discriminative training, such as contrastive learning, and benefits from fine-grained cross-modal interaction within LMMs.

### 3.3 LMMS-BASED GENERATIVE RETRIEVAL

The above proxies make it possible to calculate cross-modal similarities for T2I-R. However, it is inefficient due to $|\mathcal{G}|$ times of forward propagation, each time with the extensive parameterization and the heavy internal attention interaction. To tackle this issue and achieve an optimal balance between efficiency and recall, we introduce forward beam search and reverse re-ranking, as shown in Fig. 2.

**Forward Beam Search (FBS)**. Inspired by the advancement of generative retrieval (Tay et al., 2022; Li et al., 2024b), we adopt constraint generation via autoregressive token decoding and beam search (Freitag & Al-Onaizan, 2017) with the beam size $B$ to recall $B$ images. Specifically, we compress all images in the database into discrete tokens and store them in a Trie structure. This Trie structure constrains the sampling space and ensures that the generated prefix at any timestep corresponds to at least one image in the database. Once the beam search is finished, we could obtain a ranking list of $B$ sequences of visual tokens, each of which corresponds to an image. This process aims to obtain a list of images given the prompt, thus the direction has to be $X \rightarrow Y$, which means we can only adopt the two forward proxies illustrated in Eqn. 2 or Eqn. 3. However, FBS method could significantly improve the efficiency since it only requires $N(N \ll |\mathcal{G}|)$ times of forward propagation of LMMs, where $N$ denotes the length of the visual token sequence for an image.

**Reverse Re-Ranking (RRR)**. Despite the improved efficiency, the semantic matching ability of the two forward proxies is noticeably weaker than that of the reverse proxy in Eqn. 4, as shown in Tab. 1. Given the ranking list $\mathcal{R} = [Y_1, ..., Y_B]$ obtained by forward beam search, we resort to the reverse proxy in Eqn. 4 for re-ranking and attain the final ranking list $\mathcal{R}^*$.

### 3.4 TIGeR-ONE: Unifying Generation and Retrieval with One LMM

The discrete visual tokenization strategy enables LMMs to generate both language and visual content in an autoregressive generation manner. The proposed forward beam search in Sec. 3.3 performs retrieval in the same autoregressive manner to generate visual tokens (which have been tokenized and saved in a database). Naturally, we can unify generation and retrieval with one LMM under the autoregression framework and make a decision between the generated and retrieved images.

**Synchronous Generation and Retrieval**. In TIGeR-ONE, an LMM can synchronously conduct unconstrained and constrained token decoding processes for image generation and retrieval, respectively. As shown in Fig. 2, these two tasks can be performed in parallel by maintaining respective search paths, which only requires $N$ forward propagations, ensuring efficiency. Each path corresponds to a sequence of discrete visual tokens. We can generate a new image $Y^G$ by a diffusion decoder conditioned on the sequence, and meanwhile, immediately find the retrieved top-1[1] image $Y^R$.

**Decision Making**. Given the generated $Y^G$ and the retrieved $Y^R$, we choose the better one based on the discriminative ability of LMMs, as discussed in Sec. 3.3. Specifically, we calculate two similarities $s(X, Y^G)$ and $s(X, Y^R)$ using one of the three proxies and choose the image with the higher similarity. Since we have computed similarities between the given prompt and shortlisted images through forward beam search and reverse re-ranking using the debiasing and reverse proxies, the decision process incurs no additional computational cost and can be performed efficiently.

## 4 TIGeR-Bench

To evaluate the performance of our method on TIGeR, we build a comprehensive benchmark (TIGeR-Bench), as shown in Fig. 1. It covers creative and knowledge-intensive image acquisition domains.

**Creative Domains**. Creative images emphasize the intricate visual content that is challenging to capture in the real world. It includes unusual and *counterfactual compositions* of concepts (*e.g.*, "A steamed train bellows rainbow-hued smoke") and imaginary scenes aligning with *real users' preference*. To meet the two aspects, we collect prompt-image pairs from the well-designed WHOOPS! (Bitton-Guetta et al., 2023) dataset and a large-scale open dataset named Pick-a-Pic (Kirstain et al., 2024) which stems from a web platform collecting real users' creative intention.

**Knowledge-intensive Domains**. Acquiring knowledge-intensive images requires models with extensive world knowledge and the ability to align such knowledge with visual objects or concepts. We focus on six knowledge domains including *logo*, *history and news*, *landmark*, *food* (Min et al., 2023), *nature species*, *Wiki miscellaneous*, and collect text-image pairs from six high-quality datasets. Different from previous content-oriented data (Lin et al., 2014) covering common objects in daily life, the collected data requires stronger cross-modal knowledge alignment, as the texts often consist solely of concept names without any descriptions in appearance. We collect pairwise image-text data from eight domains, constructing the evaluation benchmark containing 6k data samples, with 3k for creative domains and 3k for knowledge domains. See Appendix A for more details.

## 5 Experiments

### 5.1 Datasets, Baselines, and Evaluation Metrics

We TIGeR-Bench to evaluate the unified performance, and the two widely-used benchmark datasets, *i.e.*, Flickr30K (Young et al., 2014) and MS-COCO (Lin et al., 2014), to assess the text-to-image retrieval performance. The compared baselines mainly include recent LMMs (Koh et al., 2023; Sun et al., 2023b;a; Ge et al., 2023; Jin et al., 2023) which can generate images, as well as generation and

---

[1]This work only considers one generated image and the top-1 retrieved image, but the proposed framework can also acquire more than one images batch-wise and choose the best one.

Table 2: Performance comparison on TIGeR-Bench. "Token" refers to visual tokenization during image synthesis, including continuous (Cont.) and discrete (Dist.) approaches. Entries by gray are expert models for T2I retrieval or generation, and those with a cyan background denote that an image query is first generated and then used for image-to-image retrieval. Entries with a gray background denote our methods. Methods with "+" denote agents, where models with "*" are decision models.

| Method | Size | LLM | Token | CLIP-T ↑ | CLIP-I ↑ |
|---|---|---|---|---|---|
| *Text-to-Image Generation* | | | | | |
| SDXL (Podell et al., 2023) | 2.6B | - | Cont. | 26.79 | 46.71 |
| GILL (Koh et al., 2023) | 8B | OPT-6.7B | Cont. | 14.16 | 13.72 |
| Emu (Sun et al., 2023b) | 14B | LLaMA-13B | Cont. | 22.26 | 40.78 |
| Emu 2 (Sun et al., 2023a) | 37B | LLaMA-33B | Cont. | 24.25 | 44.24 |
| DreamLLM (Dong et al., 2023) | 8B | Vicuna-7B | Cont. | 24.34 | 42.77 |
| SEED-LLaMA (Ge et al., 2023) | 8B | LLaMA-7B | Dist. | 22.00 | 43.02 |
| LaVIT (Jin et al., 2023) | 11B | LLaMA-7B | Dist. | 27.07 | 48.75 |
| *Text-to-Image Retrieval* | | | | | |
| CLIP (ViT-B/32) (Radford et al., 2021) | 151M | - | Cont. | 25.22 | 53.95 |
| SDXL (Podell et al., 2023) | 2.6B | - | Cont. | 15.41 | 35.96 |
| Emu (Sun et al., 2023b) | 14B | LLaMA-13B | Cont. | 14.44 | 34.46 |
| Emu 2 (Sun et al., 2023a) | 37B | LLaMA-33B | Cont. | 14.69 | 36.38 |
| DreamLLM (Dong et al., 2023) | 8B | Vicuna-7B | Cont. | 15.41 | 37.18 |
| SEED-LLaMA (Ge et al., 2023) | 8B | LLaMA-7B | Dist. | 14.78 | 36.93 |
| LaVIT (Jin et al., 2023) | 11B | LLaMA-7B | Dist. | 16.34 | 39.25 |
| GILL (Koh et al., 2023) | 8B | OPT-6.7B | Cont. | 10.96 | 16.30 |
| Ours (SEED-LLaMA) | 8B | LLaMA-7B | Dist. | 16.95 | 40.30 |
| Ours (LaVIT) | 11B | LLaMA-7B | Dist. | 21.30 | 50.03 |
| *Unified Text-to-Image Generation and Retrieval* | | | | | |
| GILL (Koh et al., 2023) | 8B | OPT-6.7B | Cont. | 12.12 | 15.25 |
| SDXL + CLIP + SEED-LLaMA* | 11B | LLaMA-7B | Cont. | 26.91 | 47.51 |
| SDXL + CLIP + Qwen2-VL* (Wang et al., 2024) | 11B | Qwen2-7B | Cont. | 27.91 | 60.65 |
| Ours (SEED-LLaMA) | 8B | LLaMA-7B | Dist. | 23.98 | 50.52 |
| Ours (LaVIT) | 11B | LLaMA-7B | Dist. | **28.45** | **61.37** |

Table 3: Text-to-image retrieval performance comparison on Flickr30K and MS-COCO. Entries by gray denote dense retrieval methods and others are generative retrieval methods. Entries with a gray background denote our methods.

| Method | Flickr30K (1K) | | | MS-COCO (5K) | | |
|---|---|---|---|---|---|---|
| | R@1 | R@5 | R@10 | R@1 | R@5 | R@10 |
| CLIP (ViT-B/32) (Radford et al., 2021) | 68.70 | 90.60 | 95.20 | 37.80 | 62.40 | 72.20 |
| GRACE (Structured ID) (Li et al., 2024b) | 37.40 | 59.50 | 66.20 | 16.70 | 39.20 | 50.30 |
| IRGen (Zhang et al., 2023) | 49.00 | 68.90 | 72.50 | 29.60 | 50.70 | 56.30 |
| Ours (LaVIT) | 68.84 | 82.92 | 86.44 | 44.81 | 62.61 | 68.28 |
| Ours (SEED-LLaMA) | **71.70** | **91.82** | **95.44** | **46.11** | **69.02** | **76.13** |

retrieval expert models (Podell et al., 2023; Radford et al., 2021). Following T2I-G (Podell et al., 2023; Esser et al., 2024; Saharia et al., 2022), the unified performance is measured by the CLIP score (Hessel et al., 2021) including CLIP-T for text-image alignment and CLIP-I for the alignment between the predicted image and the ground-truth image. As for T2I-R (Hessel et al., 2021; Faghri et al., 2017), we adopt the standard metric Recall at K, R@K (K=1, 5, and 10) for short. Details on datasets, baselines, and evaluation metrics are provided in Appendix A.

## 5.2 PERFORMANCE COMPARISON

**Unified Performance on TIGeR-Bench.** We compare LMM baselines and our method, reporting results on TIGeR-Bench in Tab. 2, including separate and unified tasks. The base models (*i.e.*, SEED-LLaMA and LaVIT) and Ours share the same architectures and parameters. The key differences lie in the integration of FBS, RRR, and decision-making mechanisms, which are training-free and non-parametric. The results demonstrate our method outperforms expert generation (Podell et al., 2023) or retrieval models (Radford et al., 2021), the state-of-the-art LMMs (Koh et al., 2023), and even agent methods with expert models for generation and retrieval and the strong understanding LMM, *i.e.*, Qwen2-VL Wang et al. (2024), as the decision model. Due to half of the data being sourced from knowledge domains, current generation models, *e.g.*, SDXL (Podell et al., 2023), and LMMs could not handle the unified problem well. Moreover, the proposed method could achieve impressive retrieval results, especially compared with other LMMs or SDXL. Compared with the vanilla SEED-LLaMA and LaVIT, our method significantly improves retrieval performance by dealing with the unified problem.

Table 5: Ablation study on TIGeR-Bench investigating Reverse Re-Ranking (RRR) and two decision-making strategies, *i.e.*, Forward with Eqn. 3 and Reverse with Eqn. 4. %Retr. denotes the percentage of retrieved images selected as results.

| RRR | Decision | Ours (SEED-LLaMA) | | | | Ours (LaVIT) | | | |
|---|---|---|---|---|---|---|---|---|---|
| | | CLIP-T ↑ | CLIP-I ↑ | R@1 ↑ | %Retr. | CLIP-T ↑ | CLIP-I ↑ | R@1 ↑ | %Retr. |
| ✗ | Forward | 22.63 | 49.71 | 26.80 | 42.10 | 27.19 | 49.59 | 28.13 | 4.38 |
| ✓ | Forward | 23.72 | 48.86 | 29.23 | 12.72 | 27.28 | 49.62 | 49.37 | 1.40 |
| ✗ | Reverse | **23.89** | **50.52** | 26.80 | 25.60 | 28.23 | 56.51 | 28.13 | 30.47 |
| ✓ | Reverse | 22.84 | 49.54 | 29.23 | 61.47 | **28.45** | **61.37** | 49.37 | 56.25 |

**Text-to-Image Retrieval Performance on Flickr30K (Young et al., 2014) and MS-COCO (Lin et al., 2014).** As shown in Tab. 3, we compare the proposed method with the representative dense retrieval model CLIP (Radford et al., 2021) and two generative retrieval baselines (Zhang et al., 2023; Li et al., 2024b) which have been specially trained on the two datasets. In contrast, the proposed method is training-free but achieves the best performance across all baselines. It verifies the effectiveness of the proposed generative retrieval method and demonstrates that LMMs are capable of retrieval despite the sole optimization objective of next-token prediction.

**Chat-to-Image Acquisition Performance on VisDial (Das et al., 2017).** Following GILL (Koh et al., 2023), we conducted experiments on VisDial to evaluate TIGeR-One based on LaVIT (Jin et al., 2023) and SEED-LLaMA (Ge et al., 2023) for image acquisition in multi-turn chat scenarios.

We compared our method with GLIDE (Nichol et al., 2021), SD v1.5 (Rombach et al., 2022), GILL, LaVIT and SEED-LLaMA, as shown in Tab. 4. The results demonstrate: 1) The original SEED-LLaMA exhibits the strongest multi-turn chat-to-image generation abilities, achieving the highest CLIP-I scores. 2) Our proposed unified framework significantly improves the performance of both LaVIT and SEED-LLaMA. 3) The improvement of our method becomes more pronounced with more rounds. For example, it achieves the most significant improvement in 10 rounds for SEED-LLaMA, which verifies the effectiveness of our method on complex multi-turn chat scenarios.

Table 4: Chat-to-image acquisition performance on Vis-Dial (Das et al., 2017). The CLIP-I score is used as the evaluation metric. Unlike previous approaches limited to image generation (Gen.), our method autonomously selects between generation and retrieval (Retr.).

| Method | Gen. | Retr. | 1 round | 5 rounds | 10 rounds |
|---|---|---|---|---|---|
| *Chat-to-Image Generation* | | | | | |
| GLIDE (Nichol et al., 2021) | ✓ | ✗ | 56.2 | 59.5 | 58.7 |
| SD-v1.5 (Rombach et al., 2022) | ✓ | ✗ | 55.2 | 62.9 | 62.2 |
| GILL (Koh et al., 2023) | ✓ | ✗ | 52.8 | 62.1 | 64.5 |
| LaVIT (Jin et al., 2023) | ✓ | ✗ | 46.6 | 52.3 | 59.3 |
| SEED-LLaMA (Ge et al., 2023) | ✓ | ✗ | **57.0** | 64.4 | 67.9 |
| *Unified Chat-to-Image Generation and Retrieval* | | | | | |
| Ours (LaVIT) | ✓ | ✓ | 51.7 | 60.0 | 65.8 |
| | | | +5.1 | +7.7 | +6.5 |
| Ours (SEED-LLaMA) | ✓ | ✓ | 57.0 | **65.4** | **70.9** |
| | | | +0.0 | +1.0 | +3.0 |

## 5.3 IN-DEPTH ANALYSIS

**Ablation Study on Reverse Re-Ranking (RRR) and Decision-Making for TIGeR.** We evaluate TIGeR-One based on SEED-LLaMA and LaVIT by different RRR and decision settings, and report the unified and retrieval performance as well as the retrieval percentage in Tab. 5. We have the following discussions: 1) RRR could consistently improve the retrieval performance for SEED-LLaMA and LaVIT, but may not help in unified performance for SEED-LLaMA, because unified performance is also influenced by decision strategies. 2) Compared with the forward decision with Eqn. 3, the reverse decision with Eqn 4 could enhance the unified performance in most cases for both models, which reflects the reverse decision may have stronger discriminative power across more domains. 3) Intuitively, we expect the most correctly retrieved images can be selected and the left wrong ones can be remedied by generation. However, we find that the two LMMs may suffer from a generation preference problem. Especially, LaVIT always prefers to choose generated images even though the retrieved ones are correct, as shown by the low %Retr. in the first two settings. One of the reasons may be that significant gaps exist between the pre-trained and fine-tuned image generation data and TIGeR-Bench. In all, besides the modality bias discussed in Tab. 1, the difference between the two directional ranking and decision may be attributed to the unbalance between captioning (image-to-text) and text-to-image data at the training phase of LMMs.

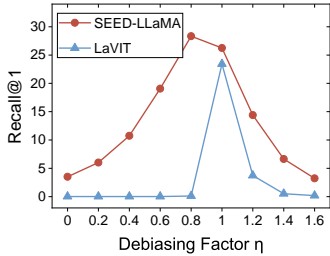 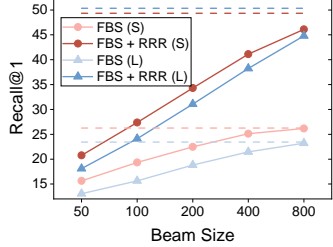 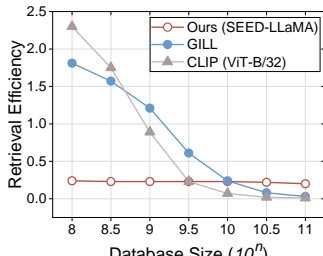

Figure 3: The influence of the debiasing factor $\eta$ in Eqn. 3 on the forward ranking performance of SEED-LLaMA and LaVIT on the MS-COCO dataset. The best performance is achieved around $\eta = 1$.

Figure 4: Retrieval performance on MS-COCO with different beam sizes and re-ranking strategies. Light and dark dash lines denote the forward and reverse ranking performance, respectively. S: SEED-LLaMA. L: LaVIT.

Figure 5: Comparison of retrieval efficiency quantified by the number of processed prompts per second among CLIP (ViT-B/32), GILL, and the proposed generative retrieval method based on SEED-LLaMA.

**Visual Modality Debiasing for Discriminative Power**. In Sec. 3.3, we discussed the visual modality bias problem with the forward $\log p(Y|X)$ in Eqn. 2 as the similarity proxy, and adopt a debiasing proxy $\log \frac{p(Y|X)}{p(Y)^\eta}$ by considering the unconditional likelihood. To explore the influence of the debiasing strength, we set different values of the factor $\eta$ in Eqn. 3 and a series of results are shown in Fig. 3. They show that the ranking performance is sensitive to the debiasing strength and reaches the highest point around $\eta = 1$, verifying the effectiveness of the unconditional debiasing strategy.

**Forward Beam Search (FBS) and Reverse Re-Ranking for Retrieval**. Considering the trade-off of retrieval efficiency and recall, we present FBS and RRR, respectively. As shown in Fig. 4, we compare the ranking (dotted lines) and retrieval (solid lines) performance and explore the impact of beam size and RRR. In the ranking experiments, we adopt the proxies in Sec. 3.3 to calculate similarities, and then rank the whole database. The comparison indicates that ranking with the debiasing proxy ($\log \frac{p(Y|X)}{p(Y)^\eta}$) seems the upper bound of FBS since FBS may miss the target image with the limited beam size. Benefiting from the reverse proxy ($\log p(X|Y)$), RRR could help FBS break through the ceiling and significantly improve recall. Additionally, regardless of similarity proxies or base LMMs employed, increasing the beam size can reduce the recall gap between retrieval and ranking.

**Efficiency of Generative Retrieval**. We analyze the efficiency of the proposed generative retrieval method for T2I-R and compare it with two dense retrieval methods, *i.e.* GILL Koh et al. (2023) and CLIP (Radford et al., 2021) in Fig. 5. The efficiency of dense retrieval gets worse with the increase in the database size due to more matching in the common feature space. In contrast, the proposed method keeps almost constant efficiency regardless of the database size.

**Prompt Expansion in Chat Scenarios.** We guide Gemini-Pro (Reid et al., 2024) and GPT-4o (OpenAI, 2024) to imagine a scenario where a user intends to know a concept and seeks an image. We provide them with detailed instructions and in-context examples, leveraging their expert language knowledge to process raw prompts. The prompts are then expanded into multi-round chat contexts, serving as input for T2I generation and retrieval.

Table 6: Performance on TIGeR-Bench (Knowledge) in *chat* scenarios, across various chat generation methods.

| Expansion Method | T2I Generation | | T2I Retrieval | | |
|---|---|---|---|---|---|
| | CLIP-T | CLIP-I | R@1 | R@5 | R@10 |
| Raw Prompt | **19.50** | 36.11 | 22.57 | 36.80 | 43.23 |
| Gemini-Pro | 17.71 | 34.58 | 17.83 | 31.70 | 36.77 |
| GPT-4o | 19.40 | **38.17** | 24.03 | 40.73 | 47.83 |

Results in Tab. 6 indicate that unifying them could utilize the abundant knowledge within LMMs to improve knowledge-intensive image acquisition.

## 5.4 QUALITATIVE ANALYSIS

In Fig. 6, we compare our methods with SDXL on TIGeR-Bench, covering both creative and knowledge domains. Besides, we explore multi-turn chat scenarios with multimodal context and both image retrieval and generation or editing requirements in Fig. 7. Owing to its training-free nature, our model-agnostic framework fully inherits the interleaved capabilities of the base model, facilitating

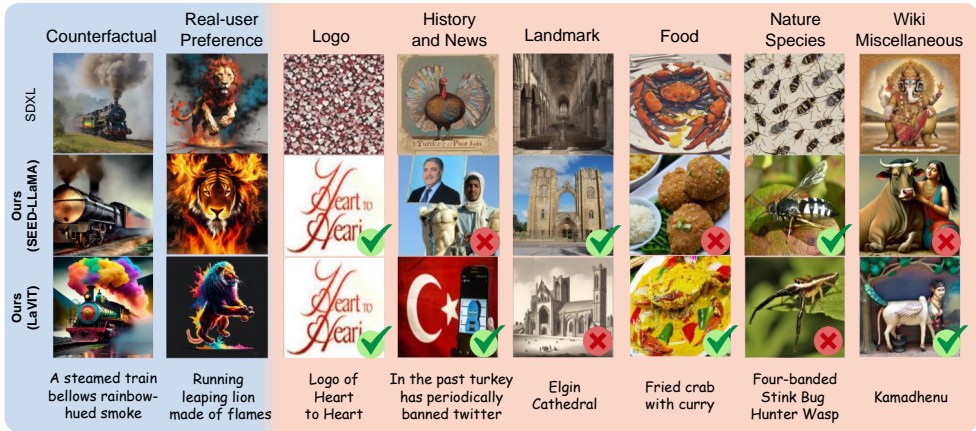

Figure 6: Qualitative results on TIGeR-Bench. The prefix prompt "Give me an image of" is omitted here. Green ticks and red crosses highlight correct and wrong retrieval results.

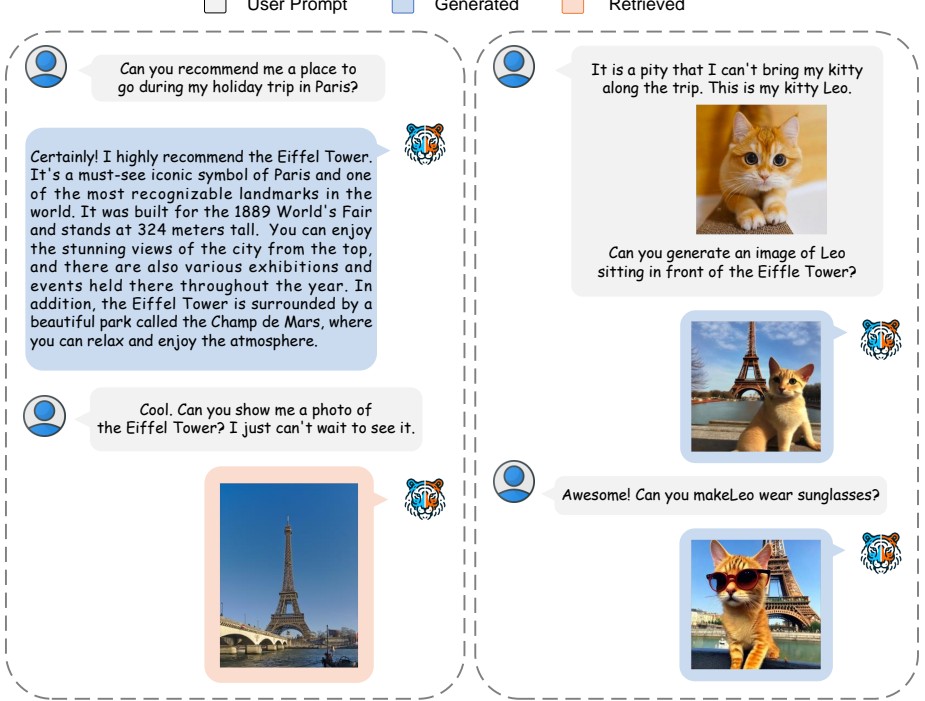

Figure 7: Example of multi-turn chat based on SEED-LLaMA with unified generation and retrieval.

accurate interleaved image generation with identity preservation. Additionally, compared to the base model, our method can proactively retrieve knowledge-intensive images (*e.g.*, the Eiffel Tower) and maintain their key characteristics throughout the interaction process.

## 6    CONCLUSION

In this work, we start with the practical requirements for image acquisition, analyze the weaknesses of single generation and retrieval, and propose to unify these two tasks within MLLMs. Toward this end, we first delve into the intrinsic discriminative abilities of MLLMs for semantic matching and propose a generative retrieval method to perform text-to-image retrieval in an auto-regressive manner. Besides, under the same auto-regressive framework, we unify generation and retrieval synchronously and present an autonomous decision strategy to select the best image. The proposed framework exhibited effectiveness and versatility across the constructed TIGeR-Bench and two retrieval benchmarks.

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

# A    TIGeR-Bench Details

## A.1    Data Collection

**Data Source.** To comprehensively evaluate unified text-to-image generation and retrieval, we build a benchmark called TIGeR-Bench, encompassing both creative and knowledge-intensive domains. For the creative domains, the data is derived from authentic user prompts that reflect real-world needs, requiring high levels of novelty and creativity. We collect the data from the WHOOPS! Bitton-Guetta et al. (2023) and Pick-a-Pic Kirstain et al. (2024) datasets:

- **WHOOPS!** Bitton-Guetta et al. (2023): The WHOOPS! dataset consists of 500 commonsense-defying prompt-image pairs created by designers. First, the designers think of counterfactual prompts by combining two elements or concepts that violate commonsense, *e.g.*, "Albert Einstein holding a smartphone". Next, they are guided to use text-to-image generation tools (*e.g.*, Midjourney, DALL-E Ramesh et al. (2021), and Stable Diffusion Rombach et al. (2022)) to synthesize images using these counterfactual prompts. Finally, the designers verify the 'weirdness' of generated images to guarantee the data quality.
- **Pick-a-Pic** Bitton-Guetta et al. (2023): The Pick-a-Pic dataset consists of real-world user prompts and corresponding generated images, annotated with user preference, gathered from the Pick-a-Pic web application. In detail, we collect our data from Pick-a-Pic v2 [2].

For knowledge-intensive domains, we collect data encompassing a wide range of categories to fulfill users' needs for visual knowledge, including Logo-2K+ Wang et al. (2020), Visual News Liu et al. (2020), Google Landmark v2 Weyand et al. (2020), Food2k Min et al. (2023), iNaturalist Van Horn et al. (2018), WIT Kirstain et al. (2024).

- **Logo-2K+** Wang et al. (2020): Logo-2K+ is a large-scale real-world logo dataset, containing 167,140 images with 2,341 categories and 10 root categories, *e.g.*, food, clothes, and institution.
- **Visual News** Liu et al. (2020): Visual News is a large-scale dataset comprising over one million news images along with associated news articles, image captions, author information, and additional metadata. Distinguished from other image captioning datasets, this dataset prioritizes factual contexts, including individuals, locations, and events, sourced from prominent news outlets such as The Guardian, BBC, USA Today, and The Washington Post.
- **Google Landmark v2** Liu et al. (2020): Google Landmark v2 includes approximately 5M images annotated with 200k distinct instance labels representing human-made and natural landmarks. It is collected from Wikimedia Commons.
- **Food2K** Min et al. (2023): Food2K is a food recognition dataset with 2,000 categories and more than 1 million images, covering cereal products, vegetables, bread, snack, soup and porridge, barbecue, egg products, dessert, beam products, seafood, fried food, and meat.
- **iNaturalist** Van Horn et al. (2018): The iNaturalist dataset is constructed to reflect the diversity of the natural world, featuring an unbalanced distribution of species. It encompasses a total of 5,000 species of plants and animals, accompanied by 859,000 images.
- **WIT** Kirstain et al. (2024): Wikipedia-based Image Text (WIT) is a large multimodal multilingual dataset, comprising 37.6 million image-text pairs representing real-world entities. It encompasses 11.5 million unique images across 108 Wikipedia languages. The texts are sourced from 3 primary channels: reference descriptions, attribution descriptions, and alt-text descriptions.

**Prompts.** The WHOOPS! and Pick-a-Pic datasets contain prompts, while Visual News also provides natural language descriptions, serving as user prompts or queries. For the remaining five datasets, only category names or concepts represented by single words or phrases are available. To address this, we utilize a template to formulate them into complete prompt sentences, *i.e.*, "Give me an image of [concept]".

These datasets, originally designed for different purposes, are effectively repurposed as the creative domain and knowledge-intensive domain candidates within the TIGeR-Bench.

---

[2]https://huggingface.co/datasets/yuvalkirstain/pickapic_v2

Table 7: The statistics of TIGeR-Bench. We keep the ratio of $1 : 1$ for creative and knowledge domains and collect 6,000 high-quality text-image pairs in total.

| Domain | Data Source | #Text-Image Pairs |
|---|---|---|
| Creative | WHOOPS! Bitton-Guetta et al. (2023) | 500 |
| | Pick-a-Pic Kirstain et al. (2024) | 2500 |
| Knowledge-intensive | Logo-2K+ Wang et al. (2020) | 500 |
| | Visual News Liu et al. (2020) | 500 |
| | Google Landmark v2 Weyand et al. (2020) | 500 |
| | Food2K Min et al. (2023) | 500 |
| | iNaturalist Van Horn et al. (2018) | 500 |
| | WIT Kirstain et al. (2024) | 500 |

## A.2 AUTOMATIC DATA FILTRATION

**Data Split**. To evaluate text-to-image generation and retrieval, we prioritize selecting the original test split of each dataset to construct TIGeR-Bench. In cases where only a validation set is provided, we default to utilizing the validation set.

**Filtration Pipeline** Given that all 8 datasets have undergone individual single-modality quality assessments during their construction, our emphasis now lies on cross-modal relevance and generation challenge properties. We proceed with the following three steps for data filtration.

1) To ensure a strong alignment between the positive text and image pairs for both generation and retrieval, we employ a filtering process to remove weakly relevant text-image pairs (*e.g.*, outliers or noisy pairs) across 7 datasets except for WHOOPS! due to its limited scale. Specifically, we calculate the CLIP-T scores ($S_{gt}$) between the ground-truth images and texts, and remove pairs with CLIP-T scores lower than 30.0. Considering the large scale of Pick-a-Pic, we then randomly sample 7,500 pairs as candidates for the following human quality validation phase.

2) As discussed in Sec. 1, T2I-G models may struggle with synthesizing knowledge-intensive images. To identify challenging concepts in the above six knowledge-intensive datasets, which pose difficulties for current state-of-the-art T2I-G models, we first employ open-sourced models including the SD series Rombach et al. (2022); Podell et al. (2023) to generate images by feeding the textual prompts in candidates as conditional input. Subsequently, we calculate the average CLIP-T scores ($S_{gen}$) over images generated by multiple models for each prompt. We then calculate the difference between the scores of the ground-truth pair and the generated pair for each prompt, *i.e.*, $\Delta = S_{gt} - S_{gen}$.

3) Finally, we select the top 1,000 unique instance pairs – comprising 1,000 different prompts and 1,000 different images – with the highest values of $\Delta$ for each knowledge dataset. The remaining examples form a new candidate set with 500 WHOOPS! instances, 7,500 Pick-a-Pic instances, and 1,000 instances for the six knowledge datasets.

## A.3 HUMAN ANNOTATOR FILTRATION

To further improve the data quality of TIGeR-Bench, human annotators were employed to mark evaluate each text-image pair across three aspects: text, image, and pair. Specifically, as for each text-image pair, considerations include the conciseness and unambiguity of the text, the clarity and usefulness of the image, and the relevance of the text-image pair. Annotators assigned a score of 0 (not satisfied) or 1 (satisfied) for each aspect. Finally, only text-image pairs meeting satisfaction across all three aspects were retained.

## A.4 DATA SAMPLING

To strike a balance between adequacy and efficiency in evaluation, we retain all 500 samples in WHOOPS!, and further randomly sample 2,500 data instances from Pick-a-Pic, along with 500 instances from each knowledge-intensive dataset. The statistics of TIGeR-Bench are presented in

Fig. 7. Maintaining a balanced ratio of $1 : 1$ between creative and knowledge domains, we finally obtain a total of 6,000 high-quality text-image pairs.

## B METHOD DETAILS

### B.1 BEAM SEARCH

Beam search Graves (2012); Boulanger-Lewandowski et al. (2013) was originally proposed for decoding tokens in sequence-to-sequence models and widely used in neural machine translation Sutskever (2014). Based on breadth-first search (BFS), it explores a search tree from the root to the leaves. At each level, beam search generates all possible child nodes based on the current prefixes, then sorts and selects the top-$B$ paths by their conditional likelihood. Unlike BFS, which considers all paths, beam search maintains only $B$ valid paths at each level and prunes others. As a result, it produces $B$ ranked sequences.

### B.2 BASE MODELS

In this work, we introduce and implement our approach for unified text-to-image generation and retrieval, based on two foundation MLLMs: SEED-LLaMA Ge et al. (2023) and LaVIT Jin et al. (2023). The details of these two models are as follows.

**SEED-LLaMA** produces 32 discrete visual codes for each image via the SEED tokenizer. This tokenizer is composed of a Causal Q-Former, a learnable codebook, and an MLP (only for training), and is trained with contrastive learning and reconstruction objectives. SEED-LLaMA takes discrete visual codes as input for multimodal understanding tasks such as image captioning and VQA, and outputs discrete visual codes. The output codes are then fed into the unCLIP-SD model Rombach et al. (2022); Ramesh et al. (2022) to generate images.

**LaVIT** obtains discrete visual codes with variable lengths using a meticulously designed dynamic visual tokenizer, which comprises a token selector, a token merger, and a reconstruction decoder (used solely for training). This tokenizer is trained with a reconstruction objective. During tokenization, LaVIT samples a binary decision mask from a Gumbel distribution to select visual patches and quantize them into discrete tokens. To ensure reproducibility and stability in tokenization, we depart from LaVIT and employ a deterministic selection method, where a patch is selected if its importance score exceeds a threshold of 0.5; otherwise, it is discarded. With this discriminative tokenization strategy, we pre-tokenize the 6 knowledge-intensive datasets of TIGeR-Bench, resulting in average, maximum, and minimum lengths of discrete tokens at 88, 130, and 37, respectively. During image generation, LaVIT first autoregressively produces a sequence of discrete visual tokens and then decodes them into an image using a diffusion model initialized with SD-v1.5 Rombach et al. (2022) or SDXL Podell et al. (2023). In contrast to SEED-LLaMA, which utilizes discrete visual tokens as input for multimodal understanding and generation, LaVIT takes continuous visual features from the token merger as input.

## C EXPERIMENTAL DETAILS

### C.1 BASELINES

As shown in Tab. 2, we compare the proposed method with several baselines on TIGeR-Bench across three aspects, *i.e.*, text-to-image generation, text-to-image retrieval, and unified generation and retrieval. We introduce these baselines in the following.

- **Text-to-Image Generation Baselines**: There include the expert model SDXL Podell et al. (2023) and recent MLLMs with image generation abilities. The MLLMs in this category are GILL Koh et al. (2023), Emu Sun et al. (2023b), Emu 2 Sun et al. (2023a), DreamLLM Dong et al. (2023), SEED-LLaMA Ge et al. (2023), and LaVIT Jin et al. (2023).

- **Text-to-Image Retrieval Baselines**: These include the expert model CLIP (ViT-B/32) Radford et al. (2021) and recent MLLMs. Currently, GILL Koh et al. (2023) is the only MLLM with retrieval ability, which maps the embeddings of special visual tokens into the CLIP

Table 8: Unified performance comparison on the *CLIP-T score* across 8 domains in TIGeR-Bench.

| Method | Creative Domains | | Logo | News | Knowledge Domains | | Food | Wiki | All |
|---|---|---|---|---|---|---|---|---|---|
| | Counterfactual | Preference | | | Landmark | Nature | | | |
| SDXL Podell et al. (2023) | 36.90 | 30.09 | 16.36 | 24.96 | 21.51 | 26.00 | 24.25 | 20.99 | 26.79 |
| CLIP (ViT-B/32) Radford et al. (2021) | 16.61 | 15.86 | 36.17 | 34.95 | 32.51 | 30.56 | 29.44 | 31.00 | 24.21 |
| GILL Koh et al. (2023) | 10.80 | 11.22 | 14.65 | 9.49 | 14.31 | 12.92 | 13.64 | 13.55 | 12.12 |
| Emu Sun et al. (2023b) | 23.61 | 23.98 | 17.24 | 19.21 | 21.54 | 23.43 | 21.25 | 21.00 | 22.26 |
| Emu 2 Sun et al. (2023a) | 29.49 | 26.21 | 23.67 | 20.85 | 19.56 | 26.09 | 20.53 | 19.81 | 24.25 |
| DreamLLM Sun et al. (2023a) | 27.16 | 23.47 | 25.57 | 25.13 | 24.27 | 23.31 | 20.78 | 24.20 | 23.98 |
| SEED-LLaMA Ge et al. (2023) | 27.18 | 23.97 | 16.73 | 19.66 | 19.03 | 22.66 | 19.63 | 19.29 | 22.00 |
| LaVIT Jin et al. (2023) | 34.60 | 29.07 | 16.70 | 25.17 | 24.14 | 29.59 | 25.07 | 24.26 | 27.07 |
| SEED-LLaMA (Ours) | 27.16 | 23.47 | 25.57 | 25.13 | 24.27 | 23.31 | 20.78 | 24.20 | 23.98 |
| LaVIT (Ours) | 32.05 | 25.39 | 35.87 | 24.30 | 32.38 | 31.28 | 27.59 | 31.01 | 28.45 |

feature space. Although other MLLMs Sun et al. (2023b;a); Dong et al. (2023); Ge et al. (2023); Jin et al. (2023) do not directly support text-to-image retrieval, we evaluate them through a two-step process: 1) generating an image query conditioned on the text prompt, and 2) performing nearest neighbor search for image-to-image retrieval using the CLIP (ViT-B/32) image encoder as the feature extractor and cosine similarity as the metric.

- **Unified Text-to-Image Generation and Retrieval**: GILL Koh et al. (2023) is the only baseline capable of performing both text-to-image generation and retrieval. It incorporates and trains a binary classifier to decide between generation and retrieval tasks.

## C.2    IMPLEMENTATION DETAILS

The proposed method is training-free and based on SEED-LLaMA Ge et al. (2023) and LaVIT Jin et al. (2023). We utilize the 8B version of SEED-LLaMA and load the parameters of supervised fine-tuning. For LaVIT, we employ the 11B model with SDXL as the pixel decoder. We combine all images in the 6 knowledge-intensive datasets and tokenize them into discrete tokens. Subsequently, we build the mapping between images and tokens. Based on these discrete tokens, we construct a Trie for efficient storage and constrained generation. The beam size for retrieval is set to 800, and the timestep for generation is 25.

## D    ADDITIONAL EXPERIMENTS

In this section, we carry out extensive experiments and obtain quantitative and qualitative results to explore the unified text-to-image generation and retrieval problem and the proposed MLLMs-based method.

### D.1    ADDITIONAL QUANTITATIVE RESULTS

#### D.1.1    UNIFIED PERFORMANCE COMPARISON

To broadly compare the performance of baselines and our method for unified text-to-image generation and retrieval, we report the results with the CLIP-T score and the CLIP-I score as the evaluation metrics across 8 domains in TIGeR-Bench, in Tab. 8 and Tab. 9, respective. In addition, we show the retrieval percentage of our method on 8 domains in TIGeR-Bench to understand the automatic decision in Tab. 10.

#### D.1.2    PROMPT/QUERY EXTENSION

In this section, we mainly study the influence of prompts on the retrieval and generation performance in knowledge-intensive scenarios. Toward this end, we first let SEED-LLaMA, Gemini Pro, and GPT-4o to explain the raw query with knowledge concepts from 6 domains. Subsequently, we concatenate the raw prompt and expanded ones to form new long text prompts and feed them to SEED-LLaMA. The results for text-to-image generation and retrieval are listed in Tab. 11 and Tab. 12, respectively. We can see that prompt/query expansion with strong LLMs could promote both generation and retrieval performance. Meanwhile, weak LLMs may introduce false explanations and do harm to the generation and retrieval performance.

Table 9: Unified performance comparison on the *CLIP-I score* across 8 domains in TIGeR-Bench.

| Method | Creative Domains | | Knowledge Domains | | | | | | All |
|---|---|---|---|---|---|---|---|---|---|
| | Counterfactual | Preference | Logo | News | Landmark | Nature | Food | Wiki | |
| SDXL Podell et al. (2023) | 65.91 | 55.38 | 14.21 | 42.09 | 35.43 | 44.94 | 45.60 | 35.40 | 46.71 |
| CLIP (ViT-B/32) Radford et al. (2021) | 31.33 | 26.68 | 93.55 | 91.04 | 71.54 | 75.38 | 71.39 | 75.59 | 53.60 |
| GILL Koh et al. (2023) | 15.93 | 13.61 | 20.38 | 15.96 | 14.34 | 18.25 | 15.49 | 14.52 | 15.25 |
| Emu Sun et al. (2023b) | 43.17 | 43.95 | 22.23 | 34.25 | 38.53 | 46.97 | 48.56 | 35.97 | 40.78 |
| Emu 2 Sun et al. (2023a) | 59.07 | 49.17 | 32.27 | 36.27 | 33.44 | 49.77 | 40.72 | 33.51 | 44.24 |
| DreamLLM Sun et al. (2023a) | 53.93 | 46.22 | 33.16 | 32.09 | 37.87 | 46.06 | 43.82 | 35.13 | 42.77 |
| SEED-LLaMA Ge et al. (2023) | 52.76 | 49.37 | 18.50 | 37.89 | 33.78 | 45.46 | 46.55 | 34.50 | 43.02 |
| LaVIT Jin et al. (2023) | 65.79 | 53.64 | 20.87 | 42.20 | 39.77 | 53.66 | 52.48 | 42.05 | 48.75 |
| SEED-LLaMA (Ours) | 52.67 | 47.81 | 51.94 | 56.86 | 48.76 | 51.99 | 52.53 | 52.52 | 50.52 |
| LaVIT (Ours) | 60.80 | 46.56 | 92.68 | 57.32 | 72.88 | 74.70 | 69.79 | 75.44 | 61.37 |

Table 10: Retrieval percentage of our method based on SEED-LLaMA and LaVIT on 8 domains in TIGeR-Bench.

| Method | Creative Domains | | Knowledge Domains | | | | | | All |
|---|---|---|---|---|---|---|---|---|---|
| | Counterfactual | Preference | Logo | News | Landmark | Nature | Food | Wiki | |
| SEED-LLaMA (Ours) | 1.0% | 9.6% | 44.2% | 44.2% | 53.6% | 26.2% | 40.2% | 49.8% | 25.6% |
| LaVIT (Ours) | 15.0% | 27.5% | 99.8% | 82.0% | 88.6% | 80.6% | 87.6% | 84.0% | 56.3% |

Table 11: Text-to-image *generation* performance on TIGeR-Bench (Knowledge) in *long text* scenarios, with CLIP-T and CLIP-I scores as evaluation metrics across various prompt/query expansion methods including Self-Expansion, Gemini Pro, and GPT-4o. For expansion, we guide LLMs to explain the appearance characteristics in detail with their expert language knowledge for given raw queries by giving them detailed instructions. After that, the queries can be expanded into longer texts and are combined with raw queries as input for text-to-image generation. We perform generative retrieval with 200 beams.

| Expansion Method | Ours (SEED-LLaMA) | | Ours (LaVIT) | |
|---|---|---|---|---|
| | CLIP-T | CLIP-I | CLIP-T | CLIP-I |
| Raw Query | 19.50 | 36.11 | **24.16** | 41.84 |
| Self-Expansion | 18.07 | 35.12 | - | - |
| Gemini-Pro | 17.48 | 34.74 | 22.38 | 39.28 |
| GPT-4o | **20.36** | **38.95** | 23.91 | **42.59** |

Table 12: Text-to-image *retrieval* performance on TIGeR-Bench (Knowledge) in *long text* scenarios, with recall as the evaluation metric across various prompt/query expansion methods including Self-Expansion, Gemini Pro, and GPT-4o. For expansion, we guide LLMs to explain the appearance characteristics in detail with their expert language knowledge for given raw queries by giving them detailed instructions. After that, the queries can be expanded into longer texts and are combined with raw queries as input for text-to-image retrieval. We perform generative retrieval with 200 beams.

| Expansion Method | Ours (SEED-LLaMA) | | | Ours (LaVIT) | | |
|---|---|---|---|---|---|---|
| | R@1 | R@5 | R@10 | R@1 | R@5 | R@10 |
| Raw Query | 22.57 | 36.80 | 43.23 | **25.63** | 43.63 | 49.40 |
| Self-Expansion | 17.20 | 30.10 | 36.77 | - | - | - |
| Gemini-Pro | 18.57 | 34.27 | 40.30 | 19.07 | 36.80 | 43.10 |
| GPT-4o | **25.00** | **42.50** | **48.90** | 25.20 | **46.03** | **52.17** |

We also carry out experiments in multimodal chat scenarios to explore the interplay between generation and retrieval, as shown in 13. Specifically, we concatenate the retrieved top-1 images behind the chat context and then evaluate the generation performance. Similarly, we concatenate the generated images behind the chat context and then evaluate the retrieval performance.

### D.1.3 ABLATION STUDY

In this part, we conduct comprehensive ablation studies on the components of the proposed method to study the effectiveness.

Table 13: Text-to-image generation and retrieval performance on TIGeR-Bench (Knowledge) in *multimodal chat* scenarios. Based on the chat contexts with pure text generated by GPT-4o, we can perform generation and retrieval. Afterwards, we concatenate the generated or retrieved top-1 images with the chat contexts and form the multimodal context, to explore the influence on retrieval and generation, respectively. Considering that LaVIT was not fine-tuned by chat instructions, we only carry out experiments based on SEED-LLaMA. We perform generative retrieval with 200 beams.

| Expansion Method | Image Context | Text-to-Image Generation | | Text-to-Image Retrieval | | |
|---|---|---|---|---|---|---|
| | | CLIP-T | CLIP-I | R@1 | R@5 | R@10 |
| Raw Query | - | **19.50** | 36.11 | **22.57** | **36.80** | **43.23** |
| GPT-4o | Retrieved | 18.62 | **38.20** | - | - | - |
| GPT-4o | Generated | - | - | 15.87 | 29.13 | 35.60 |

Table 14: Comparison of *CLIP-T score* between the unified method and single generation and retrieval variants based on SEED-LLaMA and LaVIT on 8 domains of TIGeR.

| Method | Creative Domains | | Knowledge Domains | | | | | | All |
|---|---|---|---|---|---|---|---|---|---|
| | Counterfactual | Preference | Logo | News | Landmark | Nature | Food | Wiki | |
| *Ours (SEED-LLaMA)* | | | | | | | | | |
| Generation | **27.18** | **23.97** | 16.73 | 19.66 | 19.03 | 22.66 | 19.63 | 19.29 | 22.00 |
| Retrieval | 11.04 | 10.36 | **27.87** | **26.21** | 23.73 | 20.25 | 19.56 | 22.99 | 16.95 |
| Unified | 27.16 | 23.47 | 25.57 | 25.13 | **24.27** | **23.31** | **20.78** | **24.20** | **23.98** |
| *Ours (LaVIT)* | | | | | | | | | |
| Generation | **34.60** | **29.07** | 16.70 | **25.17** | 24.14 | 29.59 | 25.07 | 24.26 | 27.07 |
| Retrieval | 13.21 | 12.17 | 35.84 | 23.64 | **32.58** | 31.08 | **27.61** | 30.76 | 21.30 |
| Unified | 32.05 | 25.39 | **35.87** | 24.30 | 32.38 | **31.28** | 27.59 | **31.01** | **28.45** |

Table 15: Comparison of *CLIP-I score* between the unified method and single generation and retrieval variants based on SEED-LLaMA and LaVIT on 8 domains of TIGeR-Bench.

| Method | Creative Domains | | Knowledge Domains | | | | | | All |
|---|---|---|---|---|---|---|---|---|---|
| | Counterfactual | Preference | Logo | News | Landmark | Nature | Food | Wiki | |
| *Ours (SEED-LLaMA)* | | | | | | | | | |
| Generation | **27.18** | **23.97** | 16.73 | 19.66 | 19.03 | 22.66 | 19.63 | 19.29 | 22.00 |
| Retrieval | 11.04 | 10.36 | **27.87** | **26.21** | 23.73 | 20.25 | 19.56 | 22.99 | 16.95 |
| Unified | 27.16 | 23.47 | 25.57 | 25.13 | **24.27** | **23.31** | **20.78** | **24.20** | **23.98** |
| *Ours (LaVIT)* | | | | | | | | | |
| Generation | **34.60** | **29.07** | 16.70 | **25.17** | 24.14 | 29.59 | 25.07 | 24.26 | 27.07 |
| Retrieval | 13.21 | 12.17 | 35.84 | 23.64 | **32.58** | 31.08 | **27.61** | 30.76 | 21.30 |
| Unified | 32.05 | 25.39 | **35.87** | 24.30 | 32.38 | **31.28** | 27.59 | **31.01** | **28.45** |

First, we compare the alignment performance between separate generation, retrieval, and unified variants across all 8 domains. The results are listed in Tab. 14 and Tab. 15 with the CLIP-T and CLIP-I score as the evaluation protocols, respectively. Besides, although Flickr30K and MS-COCO are the general datasets describing daily common scenes, we also investigate the three variants on them, as shown in Tab. 16

Second, we further study the effects of the directions of re-ranking and decision-making, across 8 domains in Tab. 17, and 2 general datasets, *i.e.*, Flickr30K and MS-COCO, in Tab. 18.

In addition, we delve into the discriminative abilities by forward and reverse ranking methods, as well as forward beam search and reverse re-ranking in Tab. 19 on 6 knowledge-intensive domains.

Table 16: Comparison between the unified method and single generation and retrieval variants based on SEED-LLaMA and LaVIT on the Flickr30K and MS-COCO datasets. Performance is evaluated by the CLIP-T score.

| Method | Flickr30K | MS-COCO |
|---|---|---|
| *Ours (SEED-LLaMA)* | | |
| Generation | 28.65 | 27.74 |
| Retrieval | 29.86 | 28.73 |
| Unified | 30.01 | 29.09 |
| *Ours (LaVIT)* | | |
| Generation | 37.05 | 35.59 |
| Retrieval | 27.54 | 27.13 |
| Unified | 33.69 | 32.24 |

Table 17: Ablation study on 8 domains of TIGeR-Bench investigating Reverse Re-Ranking (RRR) and two decision-making strategies, *i.e.*, Forward with Eqn. 3 and Reverse with Eqn. 4. Performance is evaluated by the *CLIP-T score*.

| RRR | Decision | Creative Domains | | Logo | News | Knowledge Domains | | Food | Wiki | All |
|---|---|---|---|---|---|---|---|---|---|---|
| | | Counterfactual | Preference | | | Landmark | Nature | | | |
| | | *Ours (SEED-LLaMA)* | | | | | | | | |
| | Forward | 26.64 | 22.41 | 24.02 | 23.64 | 22.85 | 20.34 | 19.63 | 22.40 | 22.63 |
| ✓ | Forward | 27.14 | **23.95** | 23.57 | 23.18 | 22.82 | 24.57 | 20.62 | 22.93 | 23.72 |
| | Reverse | **27.16** | 23.47 | 25.57 | **25.13** | **24.27** | 23.31 | 20.78 | **24.20** | **23.98** |
| ✓ | Reverse | 26.81 | 20.19 | **27.29** | 23.22 | 24.16 | **26.04** | **21.79** | 23.84 | 22.84 |
| | | *Ours (LaVIT)* | | | | | | | | |
| | Forward | **34.59** | 29.00 | 16.84 | 25.53 | 24.74 | 29.56 | 24.93 | 25.05 | 27.19 |
| ✓ | Forward | 32.05 | **29.07** | 17.15 | 25.57 | 24.76 | 29.60 | 25.17 | 25.20 | 27.28 |
| | Reverse | 33.83 | 28.17 | 24.67 | **28.88** | 28.06 | 29.02 | 25.65 | 27.84 | 28.23 |
| ✓ | Reverse | 32.05 | 25.39 | **35.87** | 24.30 | **32.38** | **31.28** | **27.59** | **31.01** | **28.45** |

Table 18: Ablation study on Flickr30K and MSCOCO investigating Reverse Re-Ranking (RRR) and two decision-making strategies, *i.e.*, Forward with Eqn. 3 and Reverse with Eqn. 4. %Retr. denotes the percentage of retrieved images selected as results.

| RRR | Decision | Flickr30K | | | MS-COCO | | |
|---|---|---|---|---|---|---|---|
| | | CLIP-T $\uparrow$ | R@1 $\uparrow$ | %Retr. | CLIP-T $\uparrow$ | R@1 $\uparrow$ | %Retr. |
| | | *Ours (SEED-LLaMA)* | | | | | |
| | Forward | 28.89 | 58.50 | 39.52 | 25.95 | 26.17 | 67.61 |
| ✓ | Forward | 29.68 | 71.70 | 26.92 | 28.71 | 46.11 | 33.91 |
| | Reverse | 30.01 | 58.50 | 35.98 | 28.61 | 26.17 | 26.23 |
| ✓ | Reverse | **30.02** | 71.70 | 51.88 | **29.09** | 46.11 | 60.69 |
| | | *Ours (LaVIT)* | | | | | |
| | Forward | 37.03 | 47.86 | 0.20 | 35.34 | 23.20 | 3.44 |
| ✓ | Forward | **37.04** | 68.84 | 0.10 | **35.58** | 44.81 | 0.23 |
| | Reverse | 36.18 | 47.86 | 24.34 | 34.67 | 23.20 | 20.60 |
| ✓ | Reverse | 33.69 | 68.84 | 41.84 | 32.24 | 44.81 | 49.11 |

## D.2 ADDITIONAL QUALITATIVE RESULTS

We showcase more examples of our SEED-LLaMA and LaVIT in both creative and knowledge-intensive domains in Fig. 8 and Fig. 9.

In the creative domain, the CLIP model, limited to retrieving images from the database, shows significant discrepancies when compared to the ground truth images. Our SEED-LLaMA and LaVIT, capable of both generation and retrieval, tend to favor image generation in the creative domain. However, our models also exhibit decision errors. For instance, as demonstrated in the last two rows of Fig. 8, the models incorrectly selected misleading retrieved images.

Table 19: Recall@1 performance comparison of Forward Ranking, Reverse Ranking, Forward Beam Search (FBS) with different beam sizes, and BFS + Reverse Re-Ranking (RRR). Experiments are conducted based on SEED-LLaMA and LaVIT on 6 knowledge-intensive domains of TIGeR-Bench.

| Method | Logo | News | Landmark | Nature | Food | Wiki | ALL |
|---|---|---|---|---|---|---|---|
| *Ours (SEED-LLaMA)* | | | | | | | |
| Forward Ranking | 56.00 | 45.60 | 22.60 | 2.60 | 4.40 | 30.40 | 26.93 |
| Reverse Ranking | 61.80 | 40.40 | 26.60 | 15.00 | 7.00 | 32.40 | 30.53 |
| FBS (#Beam=100) | 39.40 | 29.60 | 10.40 | 8.80 | 4.60 | 19.40 | 18.70 |
| FBS (#Beam=800) | 56.00 | 46.80 | 20.20 | 3.60 | 4.60 | 29.60 | 26.80 |
| FBS (#Beam=100) + RRR | 37.40 | 25.80 | 10.00 | 10.20 | 6.00 | 18.60 | 18.00 |
| FBS (#Beam=800) + RRR | 61.20 | 39.60 | 22.60 | 15.40 | 6.80 | 29.80 | 29.23 |
| *Ours (LaVIT)* | | | | | | | |
| Forward Ranking | 37.80 | 52.20 | 25.00 | 10.00 | 11.40 | 33.20 | 28.27 |
| Reverse Ranking | 92.20 | 41.00 | 56.00 | 36.40 | 23.60 | 63.80 | 52.17 |
| FBS (#Beam=100) | 16.60 | 46.00 | 20.20 | 9.40 | 10.00 | 28.40 | 21.77 |
| FBS (#Beam=800) | 37.00 | 53.40 | 24.40 | 10.80 | 11.40 | 31.80 | 28.13 |
| FBS (#Beam=100) + RRR | 27.00 | 30.80 | 30.60 | 16.80 | 15.40 | 41.00 | 26.93 |
| FBS (#Beam=800) + RRR | 87.20 | 40.40 | 51.60 | 34.60 | 23.00 | 59.40 | 49.37 |

As shown in Fig. 8, our models has the advantages over SDXL in the knowledge-intensive domain, accurately retrieving the correct results. However, decision errors still occur. We leave further exploration of the decision strategy for future work.

In Fig. 10, we compare our models with current Text-to-Image baseline models such as Emu2, DreamLLM, and GILL, which can autonomously decide between retrieval and generation. Our models are consistently retrieving the correct images in the knowledge-intensive domain. In this domain, Emu2, DreamLLM, and GILL fail to generate closely matching images, highlighting the limitations of current MLLMs.

We further explored two scenarios: Augmented Generation for Better Retrieval and Augmented Retrieval for Better Generation. In the Augmented Generation for Better Retrieval scenario, we first use the MLLM's capability to generate an image before performing image retrieval. The generated image, along with the retrieval prompt, is then used as input for the retrieval process. As shown in Fig. 11, generating an image beforehand improves the model's retrieval performance.

In the Augmented Retrieval for Better Generation scenario, we leverage our model's generative retrieval capabilities to perform an image retrieval before generating an image. The retrieved image, along with the generation prompt, is then used as input for the generation process, similar to Retrieval-Augmented Generation (RAG). As shown in Fig. 12, performing image retrieval beforehand improves the stability and quality of the generated images.

One major limitation of the CLIP model for retrieval is its limited context length. Our model leverages the advantage of the LLM's long context length, making retrieve with longer prompts possible. As shown in Fig. 13, extending the prompt further enhances the retrieval performance of our model.

# E   FUTURE WORK

In the future, we plan to investigate the root causes of modality biases from various perspectives, including data distribution, model architecture, and optimization objectives. We will also examine the potential impacts of these biases on generative and discriminative tasks. Additionally, we aim to study more complex contexts involving interleaved multimodal content to advance comprehensive unified generation and retrieval tasks. Finally, it would be valuable to explore the deeper relationships and possible interactions between generation and retrieval (e.g., retrieval-augmented generation and generation-augmented retrieval) within the TIGeR framework.

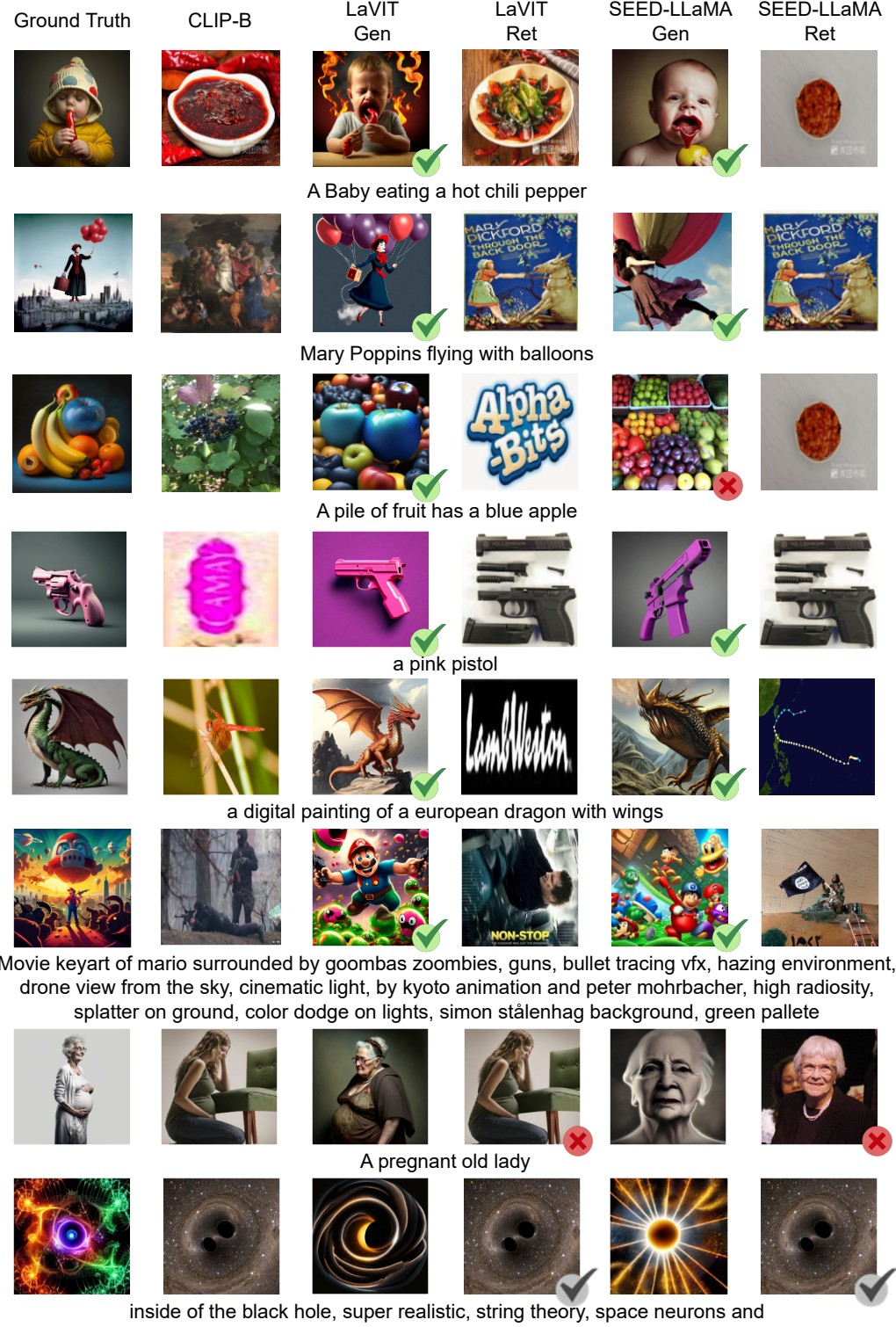

Figure 8: Qualitative results in TIGeR-Bench creative domain. We use ticks or crosses to highlight the selected results from generation or retrieval. Green ticks indicate the correct generated images and red crosses indicate the wrong retrieved images. Black ticks refer to the correct retrieved images despite the creative domain.

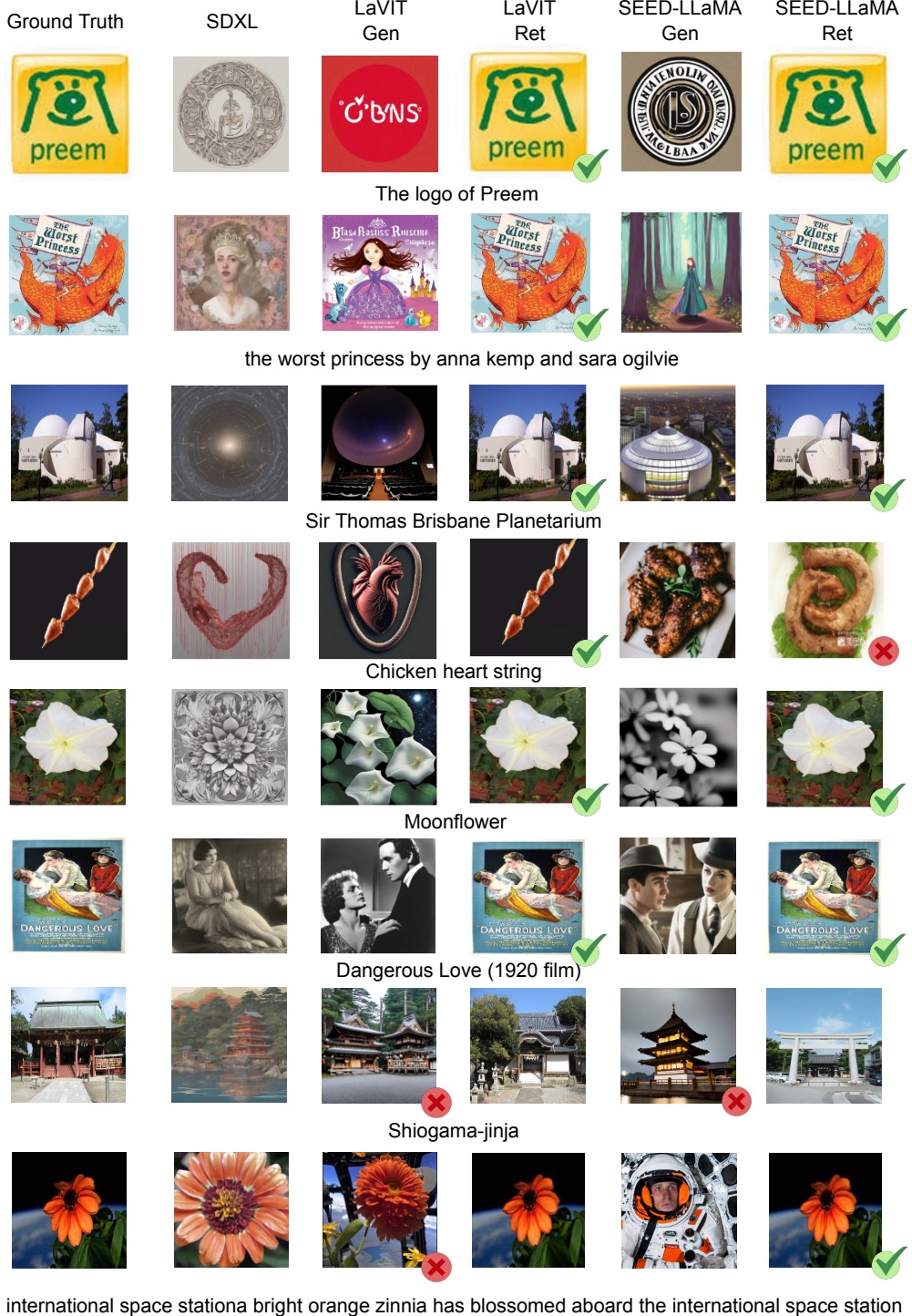

Figure 9: Qualitative results in TIGeR-Bench knowledge domain. We use ticks or crosses to highlight the selected results from generation or retrieval. Green ticks indicate the correct retrieved images and red crosses indicate the wrong generated images.

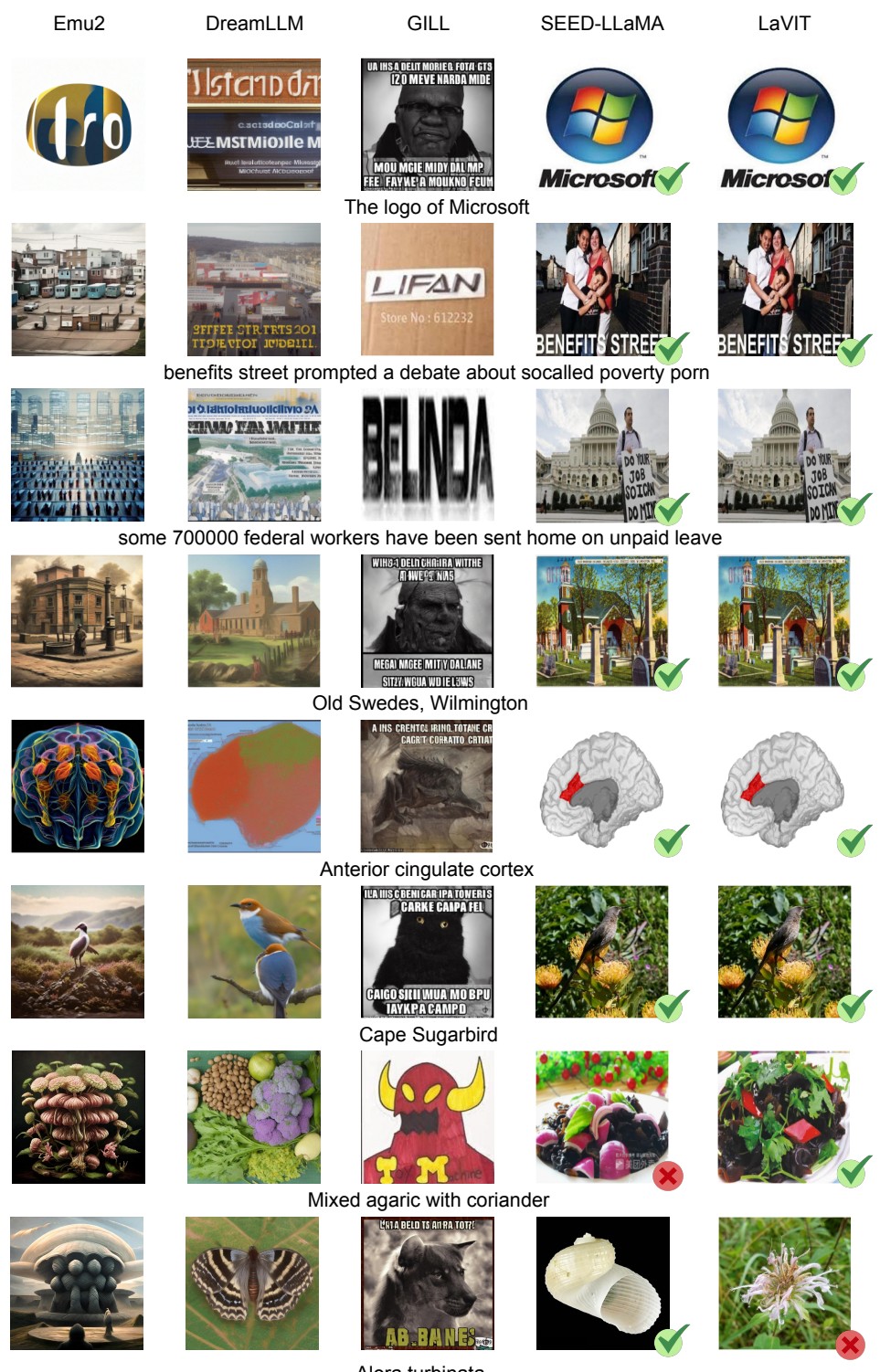

Figure 10: Qualitative results in TIGeR-Bench knowledge domain compared to Text-to-Image generation baselines. Green tick indicates the final output by our model is correct and Red cross indicates the final output by our model is incorrect.

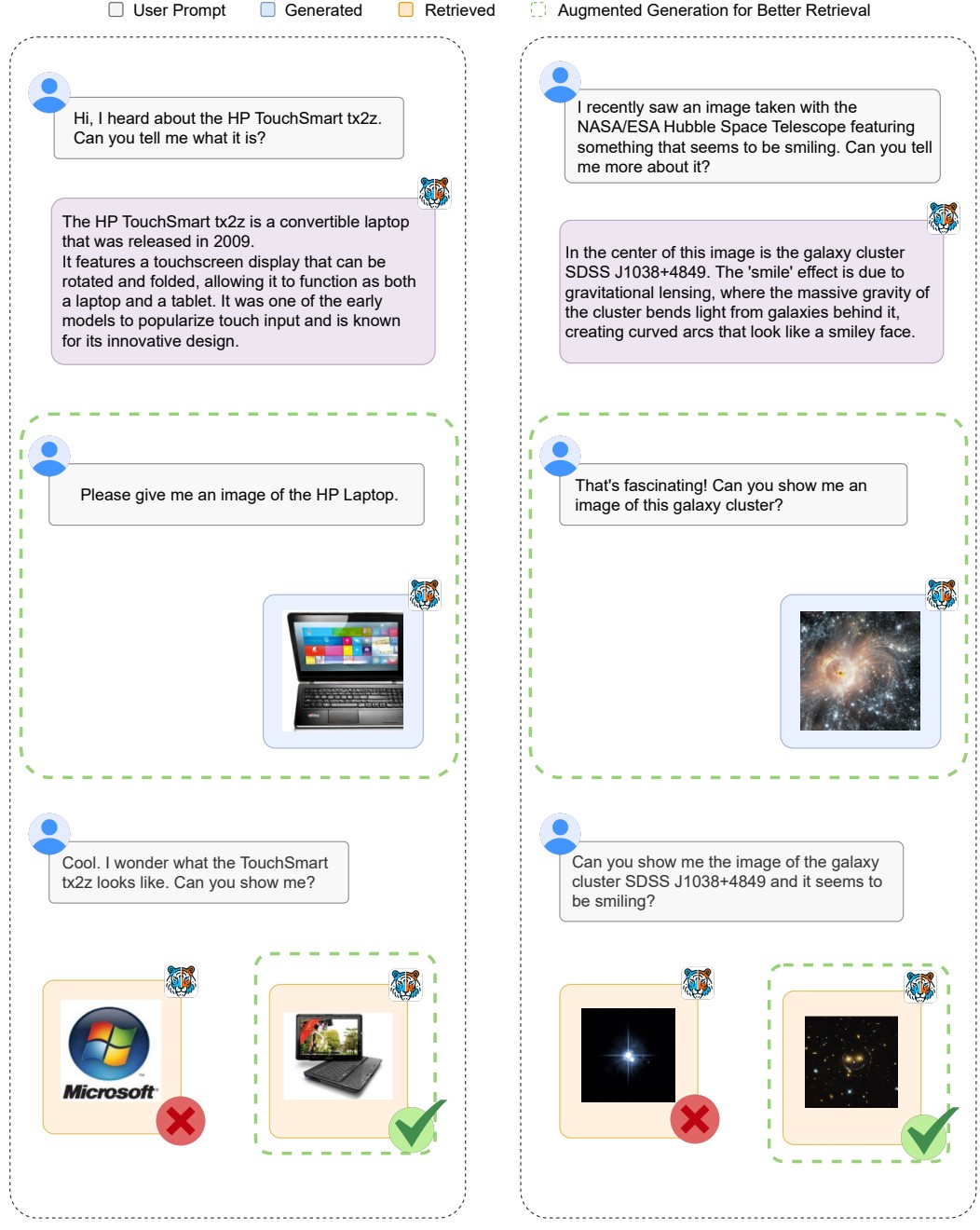

Figure 11: Augmented generation for better retrieval. Green box is the additional generation step. Green tick indicates the final output by our model is correct and Red cross indicates the fina3l output by our model is incorrect.

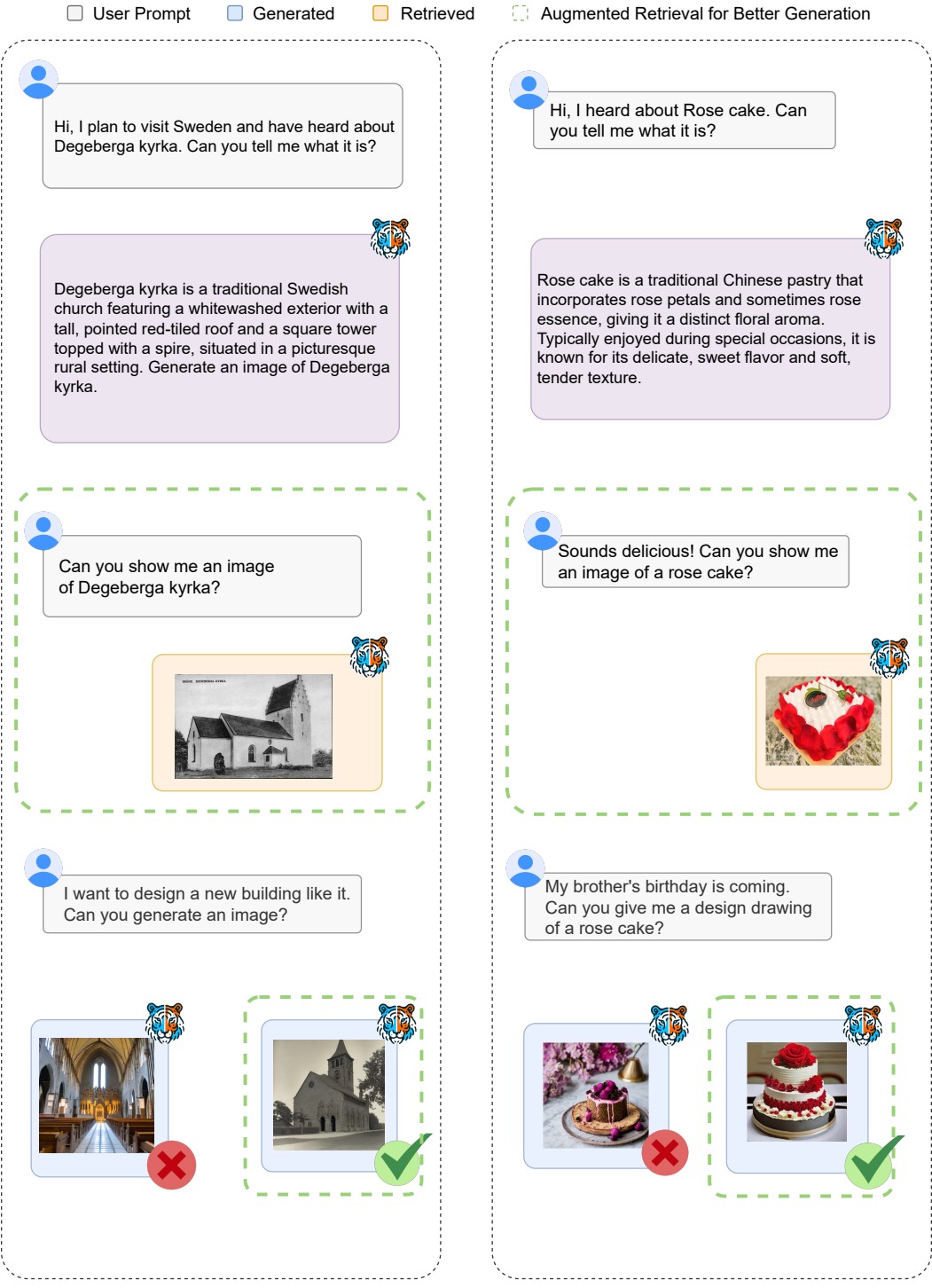

Figure 12: Augmented retrieval for better generation. Green box is the additional retrieval step. Green tick indicates the final output by our model is consistent and Red cross indicates the final output by our model is inconsistent.

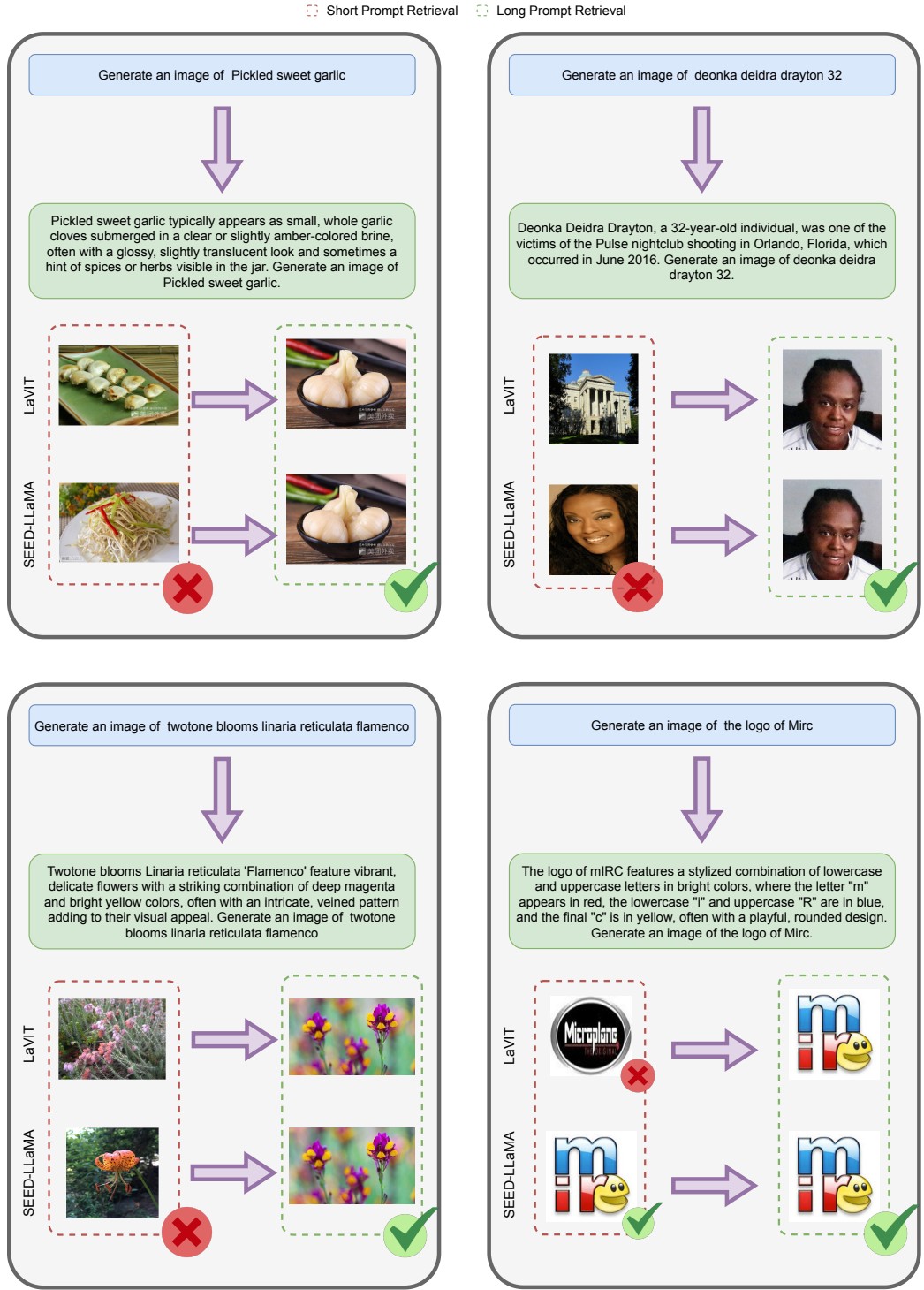

Figure 13: Short prompt and long prompt retrieval comparison on TIGeR-Bench knowledge domain. Red box is the retrieve result of short prompt. Green box is the retrieve result of long prompt. Green tick indicates the final output by our model is correct and Red cross indicates the final output by our model is incorrect.

