# OpenReview forum: "TIGeR: Unifying Text-to-Image Generation and Retrieval with Large Multimodal Models"
_ICLR.cc/2025/Conference — ICLR 2025 Poster_

### Official Review · Reviewer_wZFq · 2024-10-31

**Soundness:** 4
**Presentation:** 3
**Contribution:** 4
**Rating:** 8
**Confidence:** 4

**Summary:**

This paper rethinks the relationship between text-to-image generation and retrieval, presenting a novel and pragmatic task, i.e., TIGeR, to meet the challenging information need of humans in real-world scenarios. The authors identify limitations in traditional retrieval methods, which rely on existing image databases, and in generation approaches that struggle with knowledge-intensive content. Motivated by these limitations, the authors proposed a novel unified framework called TIGeR-ONE that combines both tasks using a single LMM. To endow LMMs with the retrieval abilities, they introduce a training-free generative retrieval method that leverages LMM's discriminative abilities, allowing for efficient text-to-image retrieval. Additionally, an autonomous decision mechanism selects either generated or retrieved images as the most suitable response to a given text prompt. The authors develop TIGeR-Bench, a benchmark covering creative and knowledge-intensive image domains, to standardize evaluation. Their approach demonstrates superior performance on the unified benchmark, text-to-image and chat-to-image retrieval and generation benchmarks.

**Strengths:**

1.	The authors introduce an innovative task that combines retrieval and generation, addressing a gap in traditional approaches that often consider these processes separately. This integrated task is highly practical, as it reflects real-world information needs where users may require either a retrieved image from a database or a newly generated one, depending on the context.
2.	The authors propose a novel unified framework that unifies text-to-image generation and retrieval within a single LMM, streamlining both tasks into a cohesive process. By using a single LMM, the authors simplify the model architecture, allowing the system to handle both retrieval of existing images and generation of new ones without additional model switching or training steps.
3.	The authors introduce a training-free generative retrieval method that enhances both the effectiveness and efficiency of text-to-image retrieval. The proposed method leverages the pre-existing discriminative abilities of Large Multimodal Models (LMMs) to perform retrieval without the need for extensive training on large datasets.
4.	They build a unified benchmark, i.e., TIGeR-Bench, to standardize evaluation across creative and knowledge-intensive tasks, serving as a strong foundation to attract more researchers to explore complex multimodal information acquisition.

**Weaknesses:**

1.	Considering there has been prior work (i.e., GILL) addressing text-to-image retrieval and generation, the difference and advantages of the proposed method over GILL should be further highlighted clearly.
2.	The paper lacks analysis on how the unified model’s performance varies with prompts that have similar intentions but are expressed differently. In practical use, different users may convey the same idea with varied wording, prompts, and questions. How sensitive is the proposed method to these variations?
3.	Given that the distribution over creative domains and knowledge domains may not be uniform in real-world scenarios, an analysis of the model’s performance and decision-making behavior under unbalanced distribution conditions would be beneficial.

**Questions:**

1.	Is there any experiment demonstrating the decision behavior of the proposed model?
2.	Compared to dense retrieval methods, such as CLIP, what advantages do MLLMs and the proposed generative retrieval framework offer?

The authors have solved my concerns.
I stand by my positive score, after reading other reviewers' comments.

---

> ### Author Response · Authors · 2024-11-23
> **Author Response to Reviewer wZFq**
>
> Thank you very much for your valuable suggestions and recognition of our contribution. We will response your concerns point by point as follows.
>
> > W1. Considering there has been prior work (i.e., GILL) addressing text-to-image retrieval and generation, the difference and advantages of the proposed method over GILL should be further highlighted clearly.
>
> The differences and advantages are summarized as the following five points:
> 1) **Task Level**: GILL focuses on equipping LLMs with image acquisition capabilities but does not address when to retrieve versus generate or how to unify them. In contrast, we analyze the strengths (e.g., generation for creativity, retrieval for knowledge-intensive tasks) and limitations (e.g., hallucination in generation, lack of creativity in retrieval) of each, highlighting the importance of unifying both paradigms.
> 2) **Model Level**: GILL aligns LLMs with external models like CLIP, making retrieval performance reliant on the external models. Our training-free approach leverages cross-modal alignment learned during pretraining and fine-tuning, enabling generative retrieval with key advantages: a) Avoids catastrophic forgetting, retaining generation capabilities. b) Resolves conflicts between next-token prediction and contrastive learning. c) Retrieval improves with generative capability enhancements.
> 3) **Decision Level**: GILL depends on a trained classifier and costly user annotations, risking overfitting with limited data. Our approach uses the intrinsic discriminative abilities to enhance decision-making without extra annotations or classifiers.
> 4) **Training Overhead**: GILL requires contrastive learning and classifier training, whereas our model is training-free.
> 5) **Inference Overhead**: As shown in Table 5, our model is more efficient for large-scale retrieval.
>
> > W2. The paper lacks analysis on how the unified model’s performance varies with prompts that have similar intentions but are expressed differently. In practical use, different users may convey the same idea with varied wording, prompts, and questions. How sensitive is the proposed method to these variations?
>
> Thank you for your thoughtful comments. We conducted experiments to analyze how diverse prompts with the same intention affect unified performance, as shown in Tables 11 and 12. We expanded raw prompts using three methods: Self-Expansion, Gemini-Pro, and GPT-4o. These expanded prompts may include useful clues, more intention, and irrelevant or noisy information.
>
> The results indicate that Self-Expansion harms retrieval and generation performance, likely due to LMM hallucination, which introduces inaccurate contexts and degrades the prompt. Conversely, GPT-4o significantly improves both tasks, demonstrating our method's potential for further gains with accurately rewritten or expanded prompts. This finding also suggests that automatic prompt engineering could enhance the proposed generative retrieval framework.
>
> > W3. Given that the distribution over creative domains and knowledge domains may not be uniform in real-world scenarios, an analysis of the model’s performance and decision-making behavior under unbalanced distribution conditions would be beneficial.
>
> > Q1. Is there any experiment demonstrating the decision behavior of the proposed model?
>
> The decision behavior across different domains is quantified in Table 10, with corresponding performance shown in Tables 8 and 9. We simulate unbalanced distributions by controllably sampling from the eight domains—for example, oversampling creative domains and undersampling knowledge domains at specific ratios. Decision-making behavior and performance can then be evaluated, and domain-specific performance is predictable from Table 10. Key observations include:
>
> 1) **Decision Patterns**: LaVIT is more likely to choose generation than SEED-LLaMA, likely due to its stronger retrieval performance. Beside some potential decision biases may exist, as the retrieval ratio remains high in creative domains.
>
> 2) **Model Flexibility**: The unified model can adaptively decide between retrieval and generation. When no suitable image exists in the database or retrieval is weak, the model opts to generate a better alternative (albeit imperfect). Compared to GILL's pre-decision strategy, TIGeR-One’s post-decision strategy is more flexible, considering both databases and intrinsic retrieval and generation capabilities.
>
> >Q2. Compared to dense retrieval methods, such as CLIP, what advantages do MLLMs and the proposed generative retrieval framework offer?
>
> Thanks for your questions. We kindly recommend referring to the Official Comment by authors titled **"Key differences between TIGeR-One and dense retrieval methods based on contrastive learning"** for more highlights of the advantages of the proposed generative retrieval framework.

---

### Official Review · Reviewer_xRkP · 2024-11-01

**Soundness:** 3
**Presentation:** 3
**Contribution:** 3
**Rating:** 6
**Confidence:** 2

**Summary:**

1. Explore the retrieval capabilities of Large Language Models (LMMs) to enable text-to-image (T2I) generation retrieval.
2. Unify generation and retrieval in an autoregressive manner.
3. Propose an autonomous decision mechanism to select the best-matched image from generated and retrieved images.
4. To standardize the evaluation of unified text-to-image generation and retrieval, we construct TIGeR-Bench, a benchmark spanning both 5. creative and knowledge-intensive domains.
6. Conduct extensive experiments on TIGeR-Bench and two retrieval benchmarks, namely Flickr30K and MS-COCO.

**Strengths:**

1. Propose a unified framework for text-to-image generation and retrieval.
2. Propose an autonomous decision mechanism to automatically select the most suitable image from the retrieved images.
3. To validate the capability of unified generation and retrieval, introduce a new benchmark, TIGeR-Bench.
4. Extensive experiments on TIGeR-Bench and two retrieval benchmarks, i.e., Flickr30K and MS-COCO, demonstrate the pipeline's capabilities.
5. The article is well-written, with clear and precise explanations.

**Weaknesses:**

1. The challenges faced by generation models in knowledge-intensive tasks are mentioned, but the proposed approach’s effectiveness in consistently addressing these challenges without hallucination remains unclear. Explicit quantitative analysis on knowledge domains might be helpful.
2. Can TIGeR-ONE adapt to user-specific preferences in the decision-making process, especially in creative tasks? If not, how could this be incorporated to enhance user experience?
3. I noticed the multi-round generation in Figure 7. Could you please explain how this process maintains consistency with the user-provided image?

**Questions:**

As shown in Weaknesses

---

> ### Author Response · Authors · 2024-11-23
> **Author Response to Reviewer xRkP**
>
> We sincerely appreciate the time and effort you dedicated to reviewing our paper. We are grateful for your insightful comments and the valuable improvements which have contributed to our paper. Please see below for a point-by-point response to your comments and concerns.
>
> > W1. The challenges faced by generation models in knowledge-intensive tasks are mentioned, but the proposed approach’s effectiveness in consistently addressing these challenges without hallucination remains unclear. Explicit quantitative analysis on knowledge domains might be helpful.
>
> Thank you for your valuable feedback and suggestions! We have conducted extensive experiments on knowledge-intensive domains, with results summarized in Tables 14 and 15. For convenience, key findings are consolidated below. The results with **CLIP-T** are shown in the following.
>
> |                  | Logo   | News   | Landmark | Nature | Food   | Wiki   | Average |
> |-----------------|:------:|:------:|:--------:|:------:|:------:|:------:|:-------:|
> | SEED-LLaMA        | 16.73  | 19.66  | 19.03    | 22.66  | 19.63  | 19.29  | 19.50   |
> | Ours (SEED-LLaMA) | **25.57**  | **25.13**  | **24.27**    | **23.31**  | **20.78**  | **24.20**  | **23.88**   |
> | LaVIT             | 16.70  | **25.17**  | 24.14    | 29.59  | 25.07  | 24.26  | 24.16   |
> | Ours (LaVIT)      | **35.87**  | 24.30  | **32.38**    | **31.28**  | **27.59**  | **31.01**  | **30.41**   |
>
> The results with **CLIP-I** are shown in the following.
> |                  | Logo   | News   | Landmark | Nature | Food   | Wiki   | Average |
> |-----------------|:------:|:------:|:--------:|:------:|:------:|:------:|:-------:|
> | SEED-LLaMA        | 18.50  | 37.89  | 33.78    | 45.46  | 46.55  | 34.50  | 36.11   |
> | Ours (SEED-LLaMA) | **51.94**  | **56.86**  | **48.76**    | **51.99**  | **52.53**  | **52.52**  | **52.43**   |
> | LaVIT             | 20.87  | 42.20  | 39.77    | 53.66  | 52.48  | 42.05  | 41.84   |
> | Ours (LaVIT)      | **92.68**  | **57.32**  | **72.88**    | **74.70**  | **69.79**  | **75.44**  | **73.80**   |
>
> The findings demonstrate that our proposed method significantly improves text-image alignment, with enhanced prompt-following abilities for image acquisition.
>
> To evaluate hallucination specially, we utilized **Gemini-Pro** via the API to assign authenticity scores from 1 to 5, with each rank clearly defined and explained. The results, shown in the following table, compare our unified model with the original generation model. The higher authenticity scores achieved by our method further validate its superiority in mitigating hallucination.
> | Base Model | Generation | Unified |
> |:----------:|:----------:|:-------:|
> | SEED-LLaMA | 3.403      | **3.915**   |
> | LaVIT      | 3.590      | **3.967**   |
>
> > W2. Can TIGeR-ONE adapt to user-specific preferences in the decision-making process, especially in creative tasks? If not, how could this be incorporated to enhance user experience?
>
> Thank you for your constructive questions! While TIGeR-ONE could incorporate user-specific preferences to enhance the user experience, this would require additional engineering and is challenging to implement during the rebuttal period. We provide the following analysis and leave further exploration for future work:
>
> a) **Explicit User Preference**: For explicit requests like “Please generate an image of…,” particularly in creative tasks, current LMMs can reliably select “generation” from {retrieval, generation, auto-decision}, leveraging the strong language understanding of their LLM backbones.
>
> b) **Implicit User Preference**: If no explicit preference is expressed, our method autonomously decides based on text-image alignment, utilizing its intrinsic discriminative capabilities.
>
> > W3. I noticed the multi-round generation in Figure 7. Could you please explain how this process maintains consistency with the user-provided image?
>
> The consistency with the user-provided image stems from the interleaved image generation capabilities of the base model, i.e., SEED-LLaMA, which ensures loyalty to the multimodal context and preserves identity. **Our model-agnostic framework fully inherits these abilities**, enabling accurate interleaved image generation with identity preservation.
>
> Additionally, compared to the base model, our method can **proactively retrieve knowledge-intensive images** (e.g., the Eiffel Tower) and maintain their key characteristics throughout the interaction process.

---

> ### Author Response · Authors · 2024-11-30
> **Follow-Up on Rebuttal**
>
> Dear Reviewer xRkP,
>
> Thank you once again for your thoughtful feedback and the time you have devoted to evaluating our submission. We have carefully addressed the concerns you previously raised and provided detailed responses in our rebuttal.
>
> We kindly remind you to review our replies and re-evaluate our work in light of the clarifications provided. If you have any additional questions or concerns, please don’t hesitate to let us know.

---

> ### Author Response · Authors · 2024-12-02
>
> Dear Reviewer xRkP,
>
> Thank you once again for your valuable feedback on our paper. To address your concerns, we have incorporated additional experiments and clarifications.
>
> As the Author-Reviewer discussion period is nearing its end, we would like to kindly ask if our responses have adequately addressed your questions. We deeply appreciate your time and thoughtful consideration.
>
> Best regards,
>
> The Authors

---

### Official Review · Reviewer_KRUZ · 2024-11-01

**Soundness:** 4
**Presentation:** 3
**Contribution:** 3
**Rating:** 8
**Confidence:** 4

**Summary:**

This paper introduces a unified framework for text-to-image generation and retrieval using LMM. It proposes an efficient method leveraging LMMs' discriminative abilities. It also presents an autonomous decision mechanism for selecting between generated and retrieved images. The authors also developed TIGeR-Bench to evaluate generation and retrieval abilities across creative and knowledge-intensive domains simultaneously.

**Strengths:**

- This paper is well-written and easy to follow.
- Efficient retrieval and accurate decision-making are crucial in unifying generation and retrieval, balancing creativity and knowledge. The proposed TIGER-One solves these two problems within one LMM is elegant.
- The proposed TIGER-One seems effective and shows superior generation results in experiments.

**Weaknesses:**

- A  preliminary introduction of beam search is recommended to add.
- The authors indicate that the similarity score is calculated by one of the three proxies. More details and discussion of the proxies chosen in decision-making are recommended.
- With the help of TIGER-One, the LMM demonstrates improved results (Tab. 2).  It’s interesting to explore the addition of dream images based on knowledge (RAG) as a third option alongside generation and retrieval in future work

**Questions:**

see the weakness part

---

> ### Author Response · Authors · 2024-11-23
> **Author Response to Reviewer KRUZ**
>
> We sincerely thank you for taking the time to provide your valuable feedback! Your review has shown us various changes and clarifications we should make to our paper. To address your concerns, we provide a detailed point-by-point response below.
>
> > W1. A preliminary introduction of beam search is recommended to add.
>
> Thank you for your valuable recommendation. The preliminary introduction to beam search is as follows:
>
> Beam search [e, f] is originally proposed to decode tokens in sequence-to-sequence models and widely used in neural machine translation [g]. It is based on breadth-first search (BFS) to explore a search tree from root to leaves. At each level, beam search generates all possible child nodes based on current prefixes, then sorts and selects the top-K paths according to their conditional likelihood, $p(childNode | prefix)$. Unlike BFS, which considers all paths, beam search keeps only K valid paths at each level and prunes others. When used in LLMs or LMMs, it produces K ranked sequences.
>
> [e] Graves Alex. Sequence transduction with recurrent neural networks. arXiv’2012
>
> [f] Boulanger-Lewandowski Nicolas et al. Audio Chord Recognition with Recurrent Neural Networks. ISMIR’2013
>
> [g] Sutskever et al. Sequence to Sequence Learning with Neural Networks. arXiv’2014
>
> We will add this preliminary introduction to the manuscript to improve the readability.
>
> > W2. The authors indicate that the similarity score is calculated by one of the three proxies. More details and discussion of the proxies chosen in decision-making are recommended.
>
> We propose three training-free proxies in Sec. 3.2. In our experiments, we mainly compare the proxy 2, i.e., $\log \frac{p(Y|X)}{p(Y)}$,  and proxy 3, i.e., $\log p(X|Y)$, considering the modality bias problem in proxy 1. $X$ and $Y$ denote text and image, respectively.
>
> Experimental results in Tab 5 and Tab 17 (In the Decision column, Forward and Reverse denote the proxy 2 and proxy 3, respectively) reveal the following observations:
>
> a) **Overall Performance**: Proxy 3 outperforms Proxy 2 on unified tasks (Table 5), demonstrating stronger discriminative abilities.
>
> b) **Domain-Specific Performance**: Proxy 2 performs better in creative domains, while Proxy 3 excels in knowledge domains (Table 17).
>
> c) **Underlying Reasons**: These differences may arise from multimodal pre-training and fine-tuning strategies. Current LMMs are optimized for understanding by modeling $\log p(X|Y)$ using diverse, knowledge-rich data, leading to the superior performance Proxy 3 in knowledge-intensive tasks. In contrast, LMMs trained for image generation by modeling $\log p(Y|X)$ with creative data enhance the discriminative abilities captured by Proxy 2 in these domains.
>
> > W3. With the help of TIGER-One, the LMM demonstrates improved results (Tab. 2). It’s interesting to explore the addition of dream images based on knowledge (RAG) as a third option alongside generation and retrieval in future work.
>
> Thank you for your insightful comments! We agree that further exploring the relationship between generation and retrieval within the unified framework, as well as settings like RAG, is both meaningful and promising. Early experiments, detailed in the Appendix, provide initial insights:
>
> 1) **Textual RAG**: As shown in Table 11, we used GPT-4o to enrich user prompts, demonstrating that richer textual context improves image generation performance.
>
> 2) **Multimodal RAG**: In Table 13, we evaluated retrieval-augmented generation using retrieved images as context. While the CLIP-I score improved, CLIP-T showed slight degradation, possibly due to current LMMs’ limited interleaved text-and-image generation capabilities. Examples in Fig. 12 illustrate that RAG helps generate images with knowledge-intensive concepts.
>
> 3) **Generation-Augmented Retrieval (GAR)**: As the inverse direction, GAR improves retrieval accuracy. Results in Table 12 and Fig. 11 show that generative models can enhance retrieval by serving as effective query expansion modules.
> We believe further exploration of RAG and GAR will offer valuable insights for the generation and retrieval communities.

---

### Official Review · Reviewer_VQ4r · 2024-11-03

**Soundness:** 2
**Presentation:** 3
**Contribution:** 3
**Rating:** 6
**Confidence:** 3

**Summary:**

This paper studies how to effectively combine the T2I generation and retrieval into a combined framework. For this task, this paper makes two folds of contribution: 1) This paper proposes a new T2I generation and retrieval pipeline, where the generation and retrieval branches run in parallel, followed by a final image ranking/selection process to decide the final output. 2) This paper proposes a benchmark to evaluate the joint T2I generation and retrieval. The benchmark contains half creative and half knowledge based image-text pairs.

**Strengths:**

- The overall presentation of this paper is good. This paper contains a lot of content which is hard to make the presentation organized and easy to follow. The authors did a good job in my opinion.
- Joint T2I generation and retrieval is a important task. The proposed benchmark set is key to make different methods devoted for the task comparable.
- The experimental results show that the proposed method has good performance than other generation or retrieval-based methods.

**Weaknesses:**

- In my opinion, the biggest drawback of this paper is that it lacks technical novelty. This work looks more like a prototype of an application than a research paper. All modules used by this paper has been used by previous papers and the combination of them contributes very limited new knowledge or insight. I understand that this work belongs to a type of papers that is more engineer than research. I put this point inside the weakness section but I am open to any discussion about the fit of this paper to iclr with authors, other reviewers, and AC if possible.
- I am confused about the comparison results in Table 2. What is the difference between the SEED-LLaMA vs Ours (SEED-LLaMA)? Does the latter one used both retrieval and generation? If so, this comparison seems to be somewhat unfair. If not, what is the components of the proposed method that lead to the performance improvement over the baseline. More discussion is welcomed.

**Questions:**

Please see the weakness section

---

> ### Author Response · Authors · 2024-11-23
> **Author Response to Reviewer VQ4r (1/2)**
>
> Thank you for taking the time to carefully review our paper! Your feedback is very valuable and insightful, and will help make our paper more clear and insightful to the reader. We present the point-to-point response as follows.
>
> > W1. In my opinion, the biggest drawback of this paper is that it lacks technical novelty. This work looks more like a prototype ...
>
> Thank you for your helpful feedback. We highlight the technical novelty of our work from the following five aspects:
>
> a) **Task for Unifying T2I Generation and Retrieval**: Previous works treat T2I generation and retrieval independently, neglecting their interrelations. In contrast, our work explores these relationships from the perspective of *generalized information acquisition*, discussing their respective advantages and disadvantages in the Introduction. We propose unifying these tasks in a single model, believing it has the potential to advance information acquisition and offer insights into the connection between understanding, generation, and even AGI.
>
> b) **Intrinsic Abilities of LMMs and Training-free Discriminative Proxies**: For the first time, we explore the discriminative abilities of LMMs, particularly text-to-image alignment. We introduce three training-free proxies compatible with LLM autoregression and next-token prediction, which provide valuable insights into modality bias and direction (e.g., text-to-image vs. image-to-text), contributing to a better understanding of LMMs and their cross-modal capabilities.
>
> c) **Training-free Generative Retrieval**: Leveraging the discriminative abilities from our proxies, we propose a training-free generative retrieval approach, integrated within the autoregressive framework. This method does not require additional discriminative training (e.g., contrastive learning) and offers a novel solution for efficient and accurate retrieval using pre-trained generative models.
>
> d) **Unified Framework for T2I Generation and Retrieval**: By aligning the autoregressive paradigms of T2I generation and retrieval, we unify both tasks in a single framework, enabling parallel execution with a single forward pass. This framework autonomously decides between generated and retrieved results, providing an elegant, effective, and efficient solution.
>
> e) **Unified Benchmark for Creative and Knowledge-intensive Domains**: To evaluate the unified performance (retrieval, generation, and decision), we introduce TIGeR-Bench, a benchmark covering 2 creative and counterfactual domains, and 6 knowledge-intensive domains.
>
> > W2.1: I am confused about the comparison results in Table 2. What is the difference between the SEED-LLaMA vs Ours (SEED-LLaMA)? Does the latter one used both retrieval and generation? If so, this comparison seems to be somewhat unfair.
>
> Thank you for your questions. We will further polish the manuscript by providing more detailed explanations in the caption and improving the clarity and intuitiveness of the table. The differences between SEED-LLaMA and Ours (SEED-LLaMA) in Tab 2 are as follows:
>
> a) SEED-LLaMA is an LMM capable of generating images but **cannot directly retrieve them** before our method is introduced. Its generation performance is shown in the first group, "Text-to-Image Generation," in Table 2.
>
> b) To compare retrieval abilities, we create a baseline by combining SEED-LLaMA with CLIP. Specifically, we first perform T2I generation using SEED-LLaMA, then use the generated image as a query for I2I retrieval with CLIP. The result is shown as "SEED-LLaMA" in the second group, "Text-to-Image Retrieval," in Table 2.
>
> We believe this provides a fair comparison, as SEED-LLaMA and Ours (SEED-LLaMA) **share the same model architecture and parameters**. The only difference is that Ours (SEED-LLaMA) incorporates generative retrieval, forward beam search, reverse re-reranking, and decision-making methods, which are **training-free and non-parametric**.

---

> > ### Author Response · Authors · 2024-11-23
> > **Author Response to Reviewer VQ4r (2/2)**
> >
> > > W2.2: what is the components of the proposed method that lead to the performance improvement over the baseline. More discussion is welcomed.
> >
> > Thanks for your question and we make the following comparisons to make the effectiveness of each component clearer.
> > The ablation study of forward beam search (FBS) and reverse re-ranking (RRR) on T2I retrieval is shown in Fig 4. We summarize the key results in the following table.
> >
> > |                 | SEED-LLaMA | w/ FBS | w/ FBS, RRR | LaVIT  | w/ FBS | w/ FBS, RRR |
> > |------------------|:------------:|:--------:|:-------------:|:--------:|:--------:|:-------------:|
> > | R@1 on Flickr30K | 20.06      | 58.50  | **71.70**       | 20.62  | 47.86  | **68.84**       |
> > | R@5 on Flickr30K | 41.42      | 84.88  | **91.82**       | 41.04  | 76.58  | **82.92**      |
> > | R@1 on MSCOCO    | 8.84       | 26.17  | **46.11**       | 9.26   | 23.20  | **44.81**       |
> > | R@5 on MSCOCO    | 23.54      | 52.88  | **69.02**       | 22.71  | 46.31  | **62.61**       |
> >
> > In Tab 2 and Tab 5, we have carried out ablation studies for each component. We extract and summarize the key results in the following two tables for SEED-LLaMA and LaVIT to make a clearer comparison.
> >
> > |       | SEED-LLaMA | w/ FBS | w/ FBS, RRR | w/ FBS, Reverse Dec. |
> > |:------:|:----------:|:------:|:-----------:|:--------------------:|
> > | CLIP-T | 22.00         | 22.63  | 23.72       | **23.89**                |
> > | CLIP-I | 43.02      | 49.71  | 48.86       | **50.52**                |
> >
> > |      | LaVIT | w/ FBS | w/ FBS, RRR | w/ FBS, RRR, Reverse Dec. |
> > |:------:|:-----:|:------:|:-----------:|:-------------------------:|
> > | CLIP-T | 27.07 | 27.19  | 28.23       | **28.45**                     |
> > | CLIP-I | 48.75 | 49.59  | 56.51       | **61.37**                     |
> >
> > The results demonstrate that:
> >
> > a) **FBS** significantly enhances retrieval performance over the base LMMs, leading to further improvements in unified performance.
> >
> > b) **RRR** with proxy 3 boosts both retrieval and unified performance, leveraging superior text-image similarity estimation and mitigating modality bias through reverse likelihood.
> >
> > c) Leveraging proxy 3, the **reverse decision** strategy autonomously selects better images, enhancing text-image alignment and improving unified performance.

---

> > > ### Comment · Reviewer_VQ4r · 2024-11-27
> > >
> > > Thank you for the detailed response. All of my concerns have been addressed.
> > >
> > > I acknowledge that I have read the author response and comments from other reviewers. I am not particularly familiar with RAG, so my evaluation is mostly based on the domain knowledge from T2I generation. I will keep my current positive rating for the following reason.
> > >
> > > * The proposed method seems to have the improved performance compared with baseline. Considering the benchmark contribution and the overall quality of the paper, this paper deserves a positive score in my opinion.
> > > * I am not convinced to give this paper a higher rating since this paper is somewhat incremental and makes limited "solid" technical contribution. I still feel this work looks more like a engineering prototype but this is more from the perspective of T2I generation rather than RAG.

---

> > > > ### Author Response · Authors · 2024-11-27
> > > > **Thanks for your response and recognition of our work**
> > > >
> > > > Thank you for your thoughtful response and recognition of our benchmark contribution and the overall quality of the paper. We deeply appreciate your time, effort, and engagement in this discussion.
> > > >
> > > > We would like to take this opportunity to further clarify and emphasize the core focus of our research: achieving training-free T2I **retrieval** and **unifying** T2I generation and retrieval within a single LMM in an autoregressive manner.
> > > >
> > > > Given that current LMMs has already possessed T2I generation capabilities, our work primarily explores four key aspects: (1) intrinsic discriminative abilities, (2) generative retrieval, (3) unification, and (4) decision-making. Moreover, our unified framework opens the door to exploring novel and exciting paradigms beyond RAG (Table 11, Figure 12, and Figure 13). For example, it supports **generation-augmented retrieval** (Table 12, Figure 11) and **interleaved retrieval and generation** (Figure 7), showcasing its versatility and potential for advancing the field.

---

### Official Review · Reviewer_x2Jm · 2024-11-05

**Soundness:** 3
**Presentation:** 2
**Contribution:** 2
**Rating:** 5
**Confidence:** 3

**Summary:**

This paper presents an approach to address the limitations of current text-to-image generation (T2I-G) and retrieval (T2I-R) systems. Particularly, it propose to unify generation and retrieval auto-regressively, and use a decision making model to decide whether to use generated or retrieved image as the response to a text prompt.

**Strengths:**

- Unification of Generation and Retrieval: The core contribution of unifying T2I-G and T2I-R within a single framework centered with LMMs. This framework leverages the strengths of both paradigms, mitigating the limitations of relying on either alone for offering visual content per user query. In an ideal world, the users can obtain a factual image generation when its query is centered at knowledge-intensive factual entity, and see a creative image when queried for imaginary scene.

- New benchmark: This paper creates a dedicated benchmark, TIGER-Bench, which accesses image generation in both creative and knowledge-intensive domains, and offers a more comprehensive evaluation platform.

**Weaknesses:**

- Lots of details related to presentation needs improvement:

1. Figure  1 is referring to many things that has not be introduced before, it is hard for people to understand forward beam-search & reverse re-ranking at the first glimpse. Many terms are not really explained in caption, or the introduction and people needs to leap to methodology section to really understand the meaning.

2. Section 3.2 is not very clear to me, I couldn't find a connection one why those three metrics are used, and also why the metric it has to be training-free. Meanwhile, I also think that comparing to those proposed proxy metrics, a more straightforward way is to directly ask the LMM a visual question, i.e. "<image> Is the presented image is aligned with the following caption? <caption>. Answer yes or no.", and then measure the likelihood on yes and no.

3. Given that the authors also agree that "It is inefficient due to |G| times of forward propagation" for computing the proposed cross-modal similarity, why not considering leveraging a contrastively trained text-to-image similarity function? The forward beam search + reverse re-ranking seems unnecessarily complicated.

- Lack of simple baseline method: It would make a lot sense to compare a simple zero-shot / fine-tuned LMM that makes a decision between whether to use the SDXL's generated image or CLIP model's retrieved image, according to a user's input query. IMO, this method would be very competitive to the proposed approach.

- The core experimental result is based on author's proposed benchmark, which is not very convincing given that it is really hard to judge the quality and validity of this newly introduced benchmark. It appears to me that the authors are being the player and judge at the same time, which is quite tricky to assess the significance of this work.

- Missing relevant works:

The following works have discussed how to leverage image retrieval for image generation and should be discussed and cited.

1. Re-Imagen: Retrieval-Augmented Text-to-Image Generator
2. KITTEN: A Knowledge-Intensive Evaluation of Image Generation on Visual Entities

**Questions:**

Please see weakness for my questions

--- Post-rebuttal comments:

I improved my scores based on authors' response.

---

> ### Author Response · Authors · 2024-11-23
> **Author Response to Reviewer x2Jm (1/3)**
>
> We sincerely thank you for your time and valuable comments. Your suggestions regarding paper revisions and experimental extensions have been invaluable in helping us refine our work. To address your concerns, we provide a detailed point-by-point response below.
> > W1. Figure 1 is referring to many things that has not be introduced before, it is hard for people to understand ...
>
> Thank you for your revision suggestions. We have updated Fig. 1 by simplifying its content and transferring some details to Fig. 2. Additionally, we have refined the corresponding caption to enhance readability and make the figure easier to understand.
>
> > W2. Section 3.2 is not very clear to me, I couldn't find a connection one ..., a more straightforward way is to directly ask the LMM ...
>
> Thank you for your constructive suggestions. We agree that the proposed VQA-based approach can serve as an alternative for calculating text-image similarities. To evaluate its potential, we conducted experiments comparing the VQA strategy with our method in terms of text-image matching performance and efficiency.
>
> a) **Matching Performance**: As shown in the following table, we try three likelihood-based approaches according to answers to estimate the alignment score and then perform text-to-image retrieval on Flickr30k.
> | Method                      | R@1    | R@5    | R@10   |
> |-----------------------------|--------|--------|--------|
> | VQA, p("yes" \| X, Y)             | 24.24  | 50.38  | 62.70  |
> | VQA, 1 / p("no" \| X, Y)          | 0.64   | 4.00   | 6.88   |
> | VQA, p("yes" \| X, Y) / p("no" \| X, Y) | 41.72  | 72.44  | 82.46  |
> | Ours                        | **71.70**  | **91.82**  | **95.44**  |
>
> Besides, we also make a comparison on the proposed TIGeR-Bench combining creative and knowledge-intensive scenarios, as shown in the following table.
> | Method | CLIP-T | CLIP-I | %Retr  |
> |--------|--------|--------|--------|
> | VQA    | 21.65  | 42.71  | 10.85  |
> | Ours   | **23.98**  | **50.52**  | 25.60  |
>
> These results demonstrate that the proposed proxy metrics are more superior than the VQA-based strategy, across multiple scenarios.
>
> b) **Efficiency**: The VQA method belongs to the one-tower framework (e.g., BLIP-2), which suffers from low efficiency. As shown in the following table, the proposed method is more efficient than it.
> |                                  | VQA for Retrieval | Ours |
> |-----------------------------------|-------------------|------|
> | Efficiency (#prompts per second) ↑ | 0.027             | **0.24** |
>
> In summary, using likelihood to estimate similarity has been studied in prior work [a, b]. The proposed proxies, such as log p(Y|X), estimate the similarities based on the likelihood of prompt X or/and image Y. In contrast, the VQA-based method estimate the similarities based on the likelihood of “yes” or “no”, e.g., log p(“yes” | X, Y). The above results show that our method is more effective compared with the VQA-based method which may suffer from instability or biases caused by question-answering instructions.
>
> [a] Bengio et al. A neural probabilistic language model. JMLR’2003
>
> [b] Lin et al. Revisiting the Role of Language Priors in Vision-Language Models. ICML’2024

---

> ### Author Response · Authors · 2024-11-23
> **Author Response to Reviewer x2Jm (2/3)**
>
> > W3.1. Given that the authors also agree that ... ,why not considering leveraging a contrastively trained text-to-image similarity function? The forward beam search + reverse re-ranking seems unnecessarily complicated.
>
> Thank you for your insightful question and comment. The drawbacks of contrastive learning methods [c] have been discussed in the Introduction (line 076 – line 080) section.
>
> Kindly refer to the Official Comment by authors titled **"Key differences between TIGeR-One and dense retrieval methods based on contrastive learning"** for more highlights, where we outline the advantages of our proposed method over approaches that leverage “a contrastively trained text-to-image similarity function”.
>
> > W3.2. Lack of simple baseline method: It would make a lot sense to compare ...
>
> Thank you for your valuable suggestions. To ensure a fair comparison among single models, we do not consider the mentioned baseline initially. Following your suggestions, we conducted additional experiments, as shown in the table below. Using SDXL and CLIP as independent generation and retrieval models, respectively, we evaluated SEED-LLaMA and a stronger model, Qwen2-VL, as decision models.
>
> | X | Gen  | Retr | Decision    | CLIP-T | CLIP-I |
> |:---:|:----:|:----:|:-----------:|:------:|:------:|
> | 1 | SDXL | -    | -           | 26.79  | 46.71  |
> | 2 | -    | CLIP | -           | 25.22  | 53.95  |
> | 3 | SDXL | CLIP | SEED-LLaMA  | 26.91  | 47.51  |
> | 4 | SDXL | CLIP | Qwen2-VL    | 27.91  | 60.65  |
> | 5 | Ours |   Ours    |   Ours           | **28.45**  | **61.37**  |
>
> The results show that with SEED-LLaMA as the decision model, CLIP-T improves, but CLIP-I underperforms compared to the single CLIP model. The stronger Qwen2-VL further enhances unified performance across both metrics, demonstrating superior discriminative abilities. However, our method still outperforms these newly added baselines, validating its effectiveness. Additionally, unlike the SDXL + CLIP + Qwen2-VL pipeline, our approach offers a unified framework capable of performing generation, retrieval, and decision simultaneously within a single LMM.

---

> > ### Author Response · Authors · 2024-11-23
> > **Author Response to Reviewer x2Jm (3/3)**
> >
> > > W3.3. The core experimental result is based on author's proposed benchmark, which is not very convincing  ...
> >
> > a) Considering the lack of prior research unifying T2I generation and retrieval, there are no established benchmarks for evaluating our method. To address this, we developed TIGeR-Bench, with construction details provided in Appendix A. As outlined in Appendix A.3, we selected 2 creative domains and 6 representative knowledge-intensive domains, randomly sampling raw data for dataset construction. To ensure quality and validity, we incorporated **human annotation filtration and verification** processes, focusing on text-image relevance, authenticity, and image quality.
> >
> > b) In addition to TIGeR-Bench, we also conduct experiments on three **public benchmarks** including Flickr30K, MS-COCO, and VisDial to verify the effectiveness of the proposed method on T2I retrieval, as shown in Tab 4, Tab 5, Tab 16, and Tab 18.
> >
> > c) We also include the commercial LMM, **Gemini-Pro, as an evaluation metric** to further validate the effectiveness of our method. The evaluations focus on three aspects: text-image relevance, image authenticity, and image quality, as summarized in the table below.
> >
> > |            | Overall Relevance | Overall Authenticity | Overall Image Quality |
> > |-------------|-------------------|----------------------|-----------------------|
> > | SDXL            | 3.821 | 3.569 | 4.096  |
> > | SEED (Retr)     | 3.13  | 3.574 | 4.185  |
> > | SEED (Unified)  | **3.352** (+0.221) | **3.666** (+0.093) | 3.986 (-0.199)  |
> > | LaVIT (Retr)    | 3.545 | 3.719 | 4.302  |
> > | LaVIT (Unified) | **4.026** (+0.481) | **3.967** (+0.248) | 4.214(-0.088)  |
> >
> > Overall, the results demonstrate that unifying generation and retrieval is significantly  effective and meaningful. The proposed unified method shows significant improvements, particularly in relevance and authenticity.
> >
> > > W3.4. The following works have discussed how to leverage image retrieval for image generation and should be discussed and cited.
> >
> > Thank you for your suggestions. We will cite and discuss them in the manuscript. Below, we compare the proposed method with the two works:
> >
> > 1. **Re-Imagen**:
> >
> > a) Re-Imagen aims to improve image generation with **RAG**, which can inject knowledge but may struggle with ensuring alignment to real-world knowledge, as injected information may be altered during generation. In contrast, our method **unifies** generation and retrieval, addressing diverse information needs.
> >
> > b) Re-Imagen is a T2I generative diffusion model, whereas our method is built on LMMs, enabling more **complex and practical settings** like long-context and multimodal context generation and retrieval, as demonstrated in Tab 4, Fig 7, and Tables 11-13.
> >
> > c) While Re-Imagen focuses on RAG, our method, benefiting from the versatility of LMMs, can also perform **RAG** (Figure 12) and **generation-augmented retrieval** (Figure 11).
> >
> > 2. **KITTEN**:
> >
> > a) The proposed method is **orthogonal** to KITTEN. KITTEN focuses on the generation performance in knowledge-intensive domains and builds a benchmark to evaluate, and it may attract more research interests to pay attention to this problem. However, it overlooks the role of retrieval. Our method also pays attention to knowledge-intensive domains, but unifies generation and retrieval from a higher level of information acquisition. The improvement of generative models in these domains would further enhance the proposed method.
> >
> > b) KITTEN highlights that state-of-the-art generative models struggle with long-tail knowledge concepts. A prior work [d] demonstrates promising retrieval performance for long-tail concepts. They can strongly **support the rationality of our work**.
> >
> > [d] Alexander Long et al. Retrieval Augmented Classification for Long-Tail Visual Recognition. CVPR’2022

---

> > > ### Comment · Reviewer_x2Jm · 2024-11-24
> > > **Thanks for your response**
> > >
> > > Thanks a lot for your response. I think most of my concerns are addressed, and will increase my score accordingly.
> > >
> > > A few bits left for the authors to address in its final manuscripts:
> > > 1. Please make sure you improve your presentation (figure, text, references) as promised.
> > > 2. Though the authors made an additional study on VQA-based metric and shown that it was not as strong as the proposed metric. I would to also like to emphasize that model-based method as such really depends on the model of choice, and the conclusion might change across model. Please be mindful and make sure you conclusion is reflecting this.
> > > 3. The simple baseline comparison is quite informative, please add it to your final experiment.

---

> > > > ### Author Response · Authors · 2024-11-28
> > > > **Paper revision and further study on VQA-based T2I retrieval methods**
> > > >
> > > > Thanks very much for your response and the further constructive comments. We address the three points as follows:
> > > >
> > > > 1. We have **revised and refined Fig 1** by removing some technical details and showing them in Fig 2 for better clarity. Besides, we have incorporated, compared, and discussed **the recommended related work in lines 118-120**, with revisions marked in blue.
> > > > 2. a) Guided by your insightful comment ``model-based method as such really depends on the model of choice``, we evaluate **four extra LMMs** capable of generating images, including **DreamLLM [h], Emu2 [i], SEED-X[j], and Emu3 [k], for VQA-based T2I retrieval**. As shown the following table, the proposed method still outperforms all the compared recent LMMs by a significant margin. We think these results could further demonstrate the superiority of the proposed method on T2I retrieval, and make the conclusion solider and more reliable.
> > > >
> > > >      |                           |    R@1    |    R@5    |   R@10    |
> > > >      | ------------------------- | :-------: | :-------: | :-------: |
> > > >      | **Ours** (SEED-LLaMA, 8B) | **71.70** | **91.82** | **95.44** |
> > > >      | VQA (SEED-LLaMA, 8B)      |   41.72   |   72.44   |   82.46   |
> > > >      | VQA (DreamLLM [h], 8B)    |   45.76   |   76.68   |   86.56   |
> > > >      | VQA (Emu2 [i], 37B)       |   43.74   |   74.56   |   84.51   |
> > > >      | VQA (SEED-X [j], 17B)     |   54.10   |   81.10   |   87.72   |
> > > >      | VQA (Emu3 [k], 8B)        |   24.91   |   50.65   |   61.75   |
> > > >
> > > > 2. b) We also investigated **the sensitivity of VQA-based methods** to question design. Four different question formats with the same meaning were tested:
> > > >
> > > >         v1: Is the image aligned with the following caption? {caption} Answer yes or no.
> > > >         v2: Is the image semantically aligned with the following caption: {caption}? Answer yes or no:
> > > >         v3: Evaluate whether the image corresponds to the caption: {caption} Respond with yes or no.
> > > >         v4: Does the image match the meaning of the caption: {caption}? Provide a yes or no response.
> > > >
> > > >     Results based on the foundation model SEED-LLaMA evaluated on the Flickr30k dataset are reported in the following table.
> > > >
> > > >      |                |    R@1    |    R@5    |   R@10    |
> > > >      | :------------: | :-------: | :-------: | :-------: |
> > > >      | VQA, Question v1 |   41.72   |   72.44   |   82.46   |
> > > >      | VQA, Question v2 |   53.96   |   81.34   |   88.28   |
> > > >      | VQA, Question v3 |   53.34   |   81.50   |   88.54   |
> > > >      | VQA, Question v4 |   53.18   |   80.58   |   87.86   |
> > > >      |      Ours      | **71.70** | **91.82** | **95.44** |
> > > >
> > > >      These results demonstrate substantial variations in retrieval performance with different formats, and highlight **the instability of VQA-based methods**. While curated question designs may yield improvements, they still fall significantly short compared to the robustness and performance of our proposed generative retrieval method.
> > > >
> > > > 3. We have **added the recommended informative baselines in Tab 2** and included a discussion of these comparisons in lines 371-374.
> > > >
> > > > [h] DreamLLM: Synergistic Multimodal Comprehension and Creation. ICLR'24
> > > >
> > > > [i] Generative Multimodal Models are In-Context Learners. CVPR'24
> > > >
> > > > [j] SEED-X: Multimodal Models with Unified Multi-granularity Comprehension and Generation. arXiv'24
> > > >
> > > > [k] Emu3: Next-Token Prediction is All You Need. arXiv'24

---

> > > > ### Author Response · Authors · 2024-11-29
> > > > **Friendly Reminder to Review Our Rebuttal**
> > > >
> > > > Dear Reviewer x2Jm,
> > > >
> > > > We sincerely thank you again for your valuable feedback and the effort you’ve dedicated to reviewing our submission. We have provided detailed responses addressing the concerns you previously raised.
> > > >
> > > > We kindly request you to take a moment to review our rebuttal and re-evaluate our work based on the clarifications and updates we have provided. If there are any further questions or concerns, please feel free to reach out—we would be more than happy to address them promptly.
> > > >
> > > > Thank you once again for your time and consideration.

---

> > > > ### Author Response · Authors · 2024-12-02
> > > >
> > > > Dear Reviewer x2Jm,
> > > >
> > > > Thank you once again for your valuable feedback on our paper. To address your concerns, we have incorporated additional experiments and clarifications.
> > > >
> > > > As the Author-Reviewer discussion period is nearing its end, we would like to kindly ask if our responses have adequately addressed your questions. We deeply appreciate your time and thoughtful consideration.
> > > >
> > > > Best regards,
> > > >
> > > > The Authors

---

### Author Response · Authors · 2024-11-23
**Key differences between TIGeR-One and dense retrieval methods based on contrastive learning**

Dear reviewers,

Thank you for the time and effort you put into providing detailed and constructive feedback. Your insights are invaluable in helping us improve the quality of our work, and we are fully committed to incorporating your suggestions into our revision process.
To address the common concern regarding the **key differences between the proposed TIGeR-One framework and dense retrieval methods** (e.g., CLIP and BLIP) based on contrastive learning, we would like to re-emphasize the following four points:

a) **Elegant Unification of Generation and Retrieval**: As illustrated in Fig. 2, our forward beam search unifies generation and retrieval within an autoregressive framework. Both tasks can be performed in parallel, yielding generated and retrieved visual tokens with only one time of auto-regression.

b) **Training-Free and LLM Compatibility**: Our training-free proxies harness intrinsic cross-modal alignment from generative pre-training with next-token prediction. Inheriting this generative manner, the beam search and ranking process does not require additional training, avoiding potential conflicts between multiple learning objectives and improving training efficiency for future fine-tuning.

c) **Inference Efficiency**: Contrastive learning methods often require significant computational overhead. Two-tower frameworks (e.g., CLIP) scale with $|\mathcal{G}|$, where $|\mathcal{G}|$ denotes the database size, while one-tower frameworks (e.g., BLIP) require  $|\mathcal{G}| \times V \times T$  operations (where V and T are visual and textual tokens). In contrast, our method scales with $N (N << |\mathcal{G}|)$, where $N$ denotes the number of visual tokens representing an image, ensuring greater efficiency.

d) **Superior Retrieval Performance**: Beyond Tab. 3, we compare retrieval performance on Flickr30K and MS-COCO between the existing dense retrieval LMM, i.e., GILL [c], and our method (see the table below). Fig. 5 further highlights our efficiency advantage. These results confirm the effectiveness and superiority of our approach in both retrieval performance and efficiency.

|                 | MSCOCO |        |        | Flickr30k |        |        |
|---|:----:|:----:|:-----------:|:------:|:------:|:------:|
|                   | R@1    | R@5    | R@10   | R@1       | R@5    | R@10   |
| GILL              | 29.10  | 53.84  | 65.26  | 58.40     | 84.48  | 90.56  |
| Ours (LaVIT)      | 44.81  | 62.61  | 68.28  | 68.84     | 82.92  | 86.44  |
| Ours (SEED-LLaMA) | **46.11**  | **69.02**  | **76.13**  | **71.70**     | **91.82**  | **95.44**  |

[c] Jing Yu Koh et al. Generating images with multimodal language models. NeurIPS’23

---

### Author Response · Authors · 2024-11-25
**Summary of PDF Revisions**

Dear Reviewers,

Thank you for your thoughtful engagement in reviewing our work and for providing valuable comments and constructive suggestions. We have carefully revised the manuscript to address your feedback. In the updated PDF, revisions are highlighted in blue for your convenience. The main updates are as follows:

1. **Figure 1** has been polished to enhance intuitiveness by removing confusing details [Reviewer x2Jm].
2. We have cited and discussed Re-Imagen and KITTEN in the **Related Work** section (lines 119–121) [Reviewer x2Jm].
3. The differences between the base models and the proposed model are now explicitly highlighted to ensure **a fair performance comparison** (lines 369–371) [Reviewer VQ4r].
4. **A preliminary introduction to beam search** has been added (lines 251–257) [Reviewer KRUZ].
5. Highlight how our model maintains **identity consistency** with the user-provided image (lines 485 - 530)[Reviewer xRkP]
6. Add the **agent baselines** in Tab 2 and corresponding discussions in lines 371–374 [Reviewer x2Jm]

 We believe these revisions address your concerns and help clarify any ambiguities or misunderstandings. As the rebuttal deadline (Nov 26, AOE) approaches, we sincerely welcome your feedback and look forward to further discussions.

Best regards,

Authors

---

### Meta-Review · Area_Chair_mhB2 · 2024-12-20

**Metareview:**

This paper presents a framework to unify text-to-image generation and retrieval by running the generation and retrieval branches, followed by a image selection process for the final output.

As reviewers pointed out, the major strengths of this paper are:
1) The idea of unifying text-to-image generation and retrieval is practically quite important, the proposed framework of using forward beam search is solid.
2) A benchmark to evaluate the joint T2I generation and retrieval is also provided.
3) The proposed TIGER-One shows superior results in experiments

The major weaknesses are:
1) The author's evaluation is specific to the method they proposed.  The evaluation can be further improved by including evaluation under more realistic scenarios.
2) As shown in authors' rebuttal, a zero-shot prompting LLM-based decision model (Qwen2-VL) that decides whether or not using generated or retrieved result seems to get only slightly worse results than the much more complicated proposed system.
3) Multiple reviewers pointed out some technical parts can be presented more clearly. The current presentation is not very easy to understand.

Overall, I think the direction of unifying text-to-image generation and retrieval is very practical and the proposed method is overall solid. So I would encourage this work and recommend "Accept as a poster".

**Additional Comments On Reviewer Discussion:**

Reviewer x2Jm asked a very important question about how simple baselines (e.g., SDXL	as generation model, CLIP as retrieval model	and some LLM like Qwen2-VL as decision model) will perform, compared to the proposed complicated method. The authors provided several baseline results, and in some of them, simple baselines perform quite well, close to the proposed method. Moreover, reviewer x2Jm also asked about more convincing experiment setting, the authors provided more benchmarks to verify the T2I retrieval for the proposed method, and moreover, Gemini-Pro is also used as an evaluation metric. These new results are informative and the authors should include them in the final version.

Reviewer VQ4r mainly asked about technical novelty and ablation study,  revieweer KRUZ mainly had concerns clarifications of some technical presentation, reviewer wZFq asked about the advantage of the proposed method compared to prior work GILL, etc. For most of these questions, I feel the authors have addressed them.

---

### Decision · Program_Chairs · 2025-01-22

Accept (Poster)